# RETHINKING REWARD MODELING IN PREFERENCE-BASED LARGE LANGUAGE MODEL ALIGNMENT

**Hao Sun,**[*] **Yunyi Shen**[*]**, Jean-Francois Ton**
University of Cambridge, Massachusetts Institute of Technology, ByteDance Research

## ABSTRACT

The Bradley-Terry (BT) model is a common and successful practice in reward modeling for Large Language Model (LLM) alignment. However, it remains unclear *why* this model — originally developed for multi-player stochastic game matching — can be adopted to convert pairwise response comparisons to reward values and make predictions. Especially given the fact that only a limited number of prompt-response pairs are sparsely compared with others. In this paper, we first establish the convergence rate of BT reward models based on deep neural networks using embeddings, providing a theoretical foundation for their use. Despite theoretically sound, we argue that the BT model is not a necessary choice from the perspective of downstream optimization, this is because a reward model only needs to preserve the correct ranking predictions through a monotonic transformation of the true reward. We highlight the critical concept of *order consistency* in reward modeling, propose a simple and straightforward upper-bound algorithm, compatible with off-the-shelf binary classifiers, as an alternative order-consistent reward modeling objective. To offer practical insights, we empirically evaluate the performance of these different reward modeling approaches across more than 12,000 experimental setups, using 6 base LLMs, 2 datasets, and diverse annotation designs that vary in quantity, quality, and pairing choices in preference annotations. [Code & Data] [A full version is available at this link]

## 1   INTRODUCTION

The alignment of Large Language Models (LLMs) is crucial for their safe and effective deployment across various applications. Current research on reinforcement learning from human feedback (RLHF) (Christiano et al., 2017; Stiennon et al., 2020; Ouyang et al., 2022; Bai et al., 2022a) has largely focused on utilizing preference-based annotations provided by humans or general-purpose LLMs Bai et al. (2022b); Lee et al. (2023); Guo et al. (2024). In general, there are two primary approaches to RLHF, namely the direct policy optimization (Rafailov et al., 2024; Zhao et al., 2023; Azar et al., 2023) that aligns LLMs with supervised learning objectives, and the alternate method that constructs a reward model to guide the LLM optimization via either supervised learning or reinforcement learning (Sun, 2023; Yuan et al., 2023; Munos et al., 2023; Li et al., 2023).

Among these strategies, the Bradley-Terry (BT) model (Bradley & Terry, 1952) is commonly employed to convert pairwise comparisons into scores and has demonstrated its success in large-scale alignment systems (Ouyang et al., 2022; Touvron et al., 2023). Despite its empirical success, the theoretical justification for using the BT model in this context remains underexplored, particularly when dealing with sparse comparisons on a limited number of prompt-response pairs. Furthermore, it is unclear how necessary it is to use the BT models, and what data format is preferred in annotation. Our work aims to address these key questions by reflecting on reward modeling in LLM alignment. The subsequent sections of this paper are structured to answer those questions:

- **Question 1**: *When the number of players is greater than the number of comparisons (as often the case in LLM alignment settings), is the use of the BT model theoretically sound, and what factors contribute to its success in practice?*
  In Section 2 we first review the usage of BT from a statistics perspective. We show that the

---

[*]HS and YS contributed equally. hs789@cam.ac.uk, yshen99@mit.edu, jeanfrancois@bytedance.com.

classical BT model can be seamlessly applied to the LLM arena cases, but falls short in reward modeling due to the extremely sparse comparison and the requirement of making predictions. We explore the regression variants of BT models (e.g., Springall, 1973; Bockenholt, 1988) and their neural network extensions, providing theoretical results on their ability to approximate true reward up to an additive constant in the context of LLM alignment.

- **Question 2**: *What are alternative approaches to reward modeling other than the BT model?*
  Since the primary objective of reward modeling is to optimize LLMs outputs by identifying the good response in inferences, learning a reward function up to any monotonic transformation should be sufficient for such an objective. This fact motivates us to propose a simple objective based on *order consistency* in Section 3. We show both the BT model and Classification models fall in this objective class. Corresponding empirical studies are conducted in Section 4.

- **Question 3**: *The conventional applications of the BT model assume randomized pairwise comparisons (e.g., randomized game matches among players). Would cross-prompt comparisons lead to more effective reward modeling?*
  Our theoretical analysis emphasizes the necessity of using regression on the embedding space to predict rewards, which hints at the possibility of different types of comparisons. Specifically, we find no inherent advantages in restricting comparisons to responses from the same prompt, as is commonly done in the literature. In Section 4, we empirically investigate the impact of cross-prompt comparisons in LLM alignment and provide insights into their potential benefits.

We defer related work discussions to Appendix A. To summarize our contributions:

1. Formally, we provide a comprehensive analysis of the application of the BT model in LLM alignment, contrasting its traditional use in multi-player arenas with the unique challenges posed in this context. We analyze the underlying rationale and offer a thorough justification for applying the BT model to LLM reward modeling.
2. Theoretically, we introduce the first asymptotic theory for neural network-based BT regression in preference-based reward modeling. Our work establishes the first risk bound for BT model reward estimation in the context of LLM alignment.
3. Practically, we propose *order consistency* as a core objective of reward modeling, demonstrating how this principle can derive both the BT model and an alternative classification-based approach. This alternative offers greater flexibility compared to the BT model, broadening its applicability.
4. Empirically, we conduct extensive experiments — covering 6 base LLMs, 2 datasets, 3 response sampling methods, 6 annotation noise levels, 3 reward model implementations, 4 annotation availability scenarios, and 5 random seeds — resulting in over 12,000 runs. These experiments demonstrate the statistical efficacy of the classification-based reward models and compare them with the BT model across diverse settings.

## 2 Rethinking the Usage of BT Models in LLM Alignment

### 2.1 Two Different BT Models — Parameter Estimation and Prediction

There are at least two distinct applications of the BT model in LLM settings: (1) LLM chatbot arenas (Chiang et al., 2024) and (2) reward modeling. While both utilize pairwise comparisons, their objectives and the volume of comparisons differ significantly. In the chatbot arena, multiple LLMs compete against one another based on human feedback through paired comparisons. Here, each LLM functions as a player, and the human-annotated preferences represent the outcome of these matches. The goal is to assign a single performance score to each LLM player. In Chiang et al. (2024) 130 models were compared across more than 1.7 million comparisons, with each model participating in over $26,000$ matches on average.

In contrast, reward modeling seeks to assign a single reward value for each *prompt-response pair*, with the added challenge of predicting rewards for unseen pairs. Typically, any given prompt-response is compared to another one only once, resulting in far fewer comparisons than in the arena setting. To understand how a single model would handle these seemingly different tasks, it is beneficial to have an overview of the BT model and their variants.

The original BT model (Bradley & Terry, 1952), aligned with the Luce-Shephard choice rule (Luce, 1959; Shepard, 1957), posits that in a simplified two-option scenario, the **probability** of selecting

option $i$ from a set $i, j$ is proportional to the utility $u(\cdot)$ assigned to that option. Formally, this can be expressed as a softmax output of the log utilities $r(\cdot)$ (Bockenholt, 1988)

$$P(i \succ j) = \frac{u(i)}{u(i) + u(j)} = \frac{\exp(r(i))}{\exp(r(i)) + \exp(r(j))} = \text{softmax}(r(i), r(j)). \qquad (1)$$

In the LLM arena setting, estimating the values of $r(\cdot)$ is sufficient to achieve the primary goal of evaluating each chatbot's performance. This aligns closely with the original objective of BT model in ranking sports teams (Bradley & Terry, 1952). Previous work has shown that, with enough pairwise competition, one can estimate these ability scores well (Ford Jr, 1957; Han et al., 2020; Wu et al., 2022) up to a constant additive factor. It is shown that to estimate $N$ scores via random pairwise comparisons, the theoretical lower bound on the number of comparisons is $O(N \log(N))$[1], while the best-known methods require $O(N \log^3(N))$ comparisons (Han et al., 2020).

In contrast, the application of the BT model to reward modeling is not as straightforward. First, the implications of using the BT model in this context are not well-defined in the literature. For instance, if each prompt-response pair is treated as a player, how do we characterize the stochastic nature of human annotations as the game results? What assumptions need to be made? The answers to these questions were not elaborated in the literature (Christiano et al., 2017; Stiennon et al., 2020; Ouyang et al., 2022; Bai et al., 2022a). In Appendix B, we provide an analysis of the underlying assumptions of applying the BT model to preference-based annotation as a game.

Additionally, estimating a separate $r(\cdot)$ for each prompt-response pair is impractical. In typical LLM alignment scenarios, we often have only $N/2$ comparisons for $N$ pairs, far below the theoretical lower bound for consistent estimation (Han et al., 2020). Furthermore, unlike the arena setting, there is no clear way to predict the score for a new, unseen pair. However, this challenge is not unique to LLM alignment; sports analysts also need to estimate a team's ability before many competitions or predict the ability of a new team. A common approach in such cases is to use features or covariates, such as team composition or funding status, to predict scores. For LLM, one could have **sentence embeddings** as such covariates.

These extensions of the BT model were explored shortly after its original introduction: Springall (1973) assumed that $r(\cdot)$ could be expressed as a linear combination of covariates. In this scenario, the problem reduces to a classic logistic regression on the difference between two sets of covariates. This allows us to predict the score for a new team or prompt-response pair based on its covariates before any comparisons are made. More complex nonlinear models such as spline models have also been explored (De Soete & Winsberg, 1993). For a more comprehensive review from a statistical perspective, we refer readers to Cattelan (2012).

In practice, reward modeling in LLM alignment often employs neural networks, with multilayer perceptrons (MLPs) being a common choice to map embeddings to scores. However, there is currently no theoretical justification for why this particular choice of model and loss function is effective for learning reward functions. From a theoretical standpoint, this model variant can be viewed as a nonparametric logistic regression problem (Bockenholt, 1988; Schmidt-Hieber, 2020) with additional structural assumptions on the network, and our analysis builds on this framework. In the following section, we establish the asymptotic theory for learning reward models using MLPs and BT loss.

### 2.2 ASYMPTOTIC THEORY ON MLP-BASED BT REGRESSION IN REWARD MODELING

In preference-based LLM alignment, we work with the dataset under the form of $\mathcal{D}_{\text{pref}} = \{(x_i, y_{1,i}, y_{2,i}, h_i)\}_{i \in [n]}$, where each tuple consists of the prompt $x_i$, the corresponding responses $y_{1,i}$ and $y_{2,i}$ sampled from the LLM $\ell$ to be aligned $y_{1,i}, y_{2,i} \sim \ell(x_i)$, and the human-annotated preference $h_i$, being 1 if $y_{1,i}$ is preferred and $-1$ otherwise.

Assume we have a *known* embedding function $\Psi(\cdot, \cdot) : \mathcal{X} \times \mathcal{Y} \mapsto [0, 1]^d$, such that there exists an unknown reward function $r : \mathbb{R}^d \mapsto \mathbb{R}$, and can be expressed as $r(\Psi(x, y))$ for all $x, y$. Without loss of generality, we assume the embeddings are scaled within the range $[0, 1]$ — otherwise, we can scale the embeddings into this range. Under this framework, reward modeling reduces to learning the function $r$. Note that under this formalism there is no need for a comparison to have the same

---

[1]Intuition behind this bound can come from the fact that, given the true underlying scores, a quicksort algorithm requires $O(N \log(N))$ comparisons on average.

prompt. We will empirically explore the effects of using cross-prompt and same-prompt in our experiment section.

Denote our reward model as $\hat{r}_\theta$, parameterized by $\theta$, when there is no confusion, we will abbreviate it as $\hat{r}$. We denote the vector of two rewards as $\hat{\boldsymbol{r}}$ and the class probability is then $\mathrm{softmax}(\hat{\boldsymbol{r}})$. Thus, training this reward model reduces to training a classifier with a cross-entropy loss, where the predicted conditional class probabilities are computed as $\mathrm{softmax}(\hat{r}(\Psi(x_1, y_1)), \hat{r}(\Psi(x_2, y_2)))$. Our theoretical analysis follows closely the work of Bos & Schmidt-Hieber (2022) and Schmidt-Hieber (2020). In this setting, we consider a special case of a model that preserves anti-symmetry: if we exchange the roles of $x_1, y_1$ with $x_2, y_2$, the reward difference becomes negative.

We demonstrate that an MLP can approximate the probability that the first pair is preferred over the second, and subsequently show that this approach enables the model to learn the underlying reward function effectively. For notational simplicity, let $\boldsymbol{\Psi}_1^{(i)}$ and $\boldsymbol{\Psi}_2^{(i)}$ represent the embeddings of the $i$-th pair, where $i = 1, \ldots, n$. Without loss of generality, we assume $\boldsymbol{\Psi}^{(i)} \in [0, 1]^d$. Let the true preference probability for the $i$-th pair be $\boldsymbol{p}_0^{(i)}$, and the predicted probability be $\hat{\boldsymbol{p}}^{(i)} = (\sigma(\hat{r}^\Delta(\boldsymbol{\Psi}_1^{(i)}, \boldsymbol{\Psi}_2^{(i)})), 1 - \sigma(\hat{r}^\Delta(\boldsymbol{\Psi}_1^{(i)}, \boldsymbol{\Psi}_2^{(i)}))) = \mathrm{softmax}(\hat{\boldsymbol{r}}(\boldsymbol{\Psi}_1^{(i)}, \boldsymbol{\Psi}_2^{(i)}))$. The preference label $\boldsymbol{h}^{(i)}$ equals $(1, 0)$ if the first response pair is preferred, and $(0, 1)$ otherwise. In this way, the BT model can be reduced to a pairwise classification problem, where the likelihood is given by:

$$\widetilde{\mathcal{L}}_{\mathrm{CE}}(\boldsymbol{p}) = -\frac{1}{n} \sum_{i=1}^n (\boldsymbol{h}^{(i)})^\top \log(\boldsymbol{p}^{(i)}), \quad \hat{\boldsymbol{p}} = \arg\min_{\boldsymbol{p} \in \mathcal{F}_\theta} \widetilde{\mathcal{L}}_{\mathrm{CE}}(\boldsymbol{p}) \tag{2}$$

It is unrealistic to assume we can find an NN that actually attends the global minimum, we denote the difference between the fitted NN and the global minimum as

$$\Delta_n(\boldsymbol{p}_0, \hat{\boldsymbol{p}}) = \mathbb{E}\left[\widetilde{\mathcal{L}}_{\mathrm{CE}}(\hat{\boldsymbol{p}}) - \min_{\boldsymbol{p} \in \mathcal{F}_\theta} \widetilde{\mathcal{L}}_{\mathrm{CE}}(\boldsymbol{p})\right] \tag{3}$$

We consider the truncated KL risk following Bos & Schmidt-Hieber (2022) to overcome the divergence problem of KL risk.

**Definition 2.1** (Truncated KL risk (Bos & Schmidt-Hieber, 2022)). The $B-$truncated KL risk for a probability estimator $\hat{\boldsymbol{p}}$

$$R_B(\boldsymbol{p}_0, \hat{\boldsymbol{p}}) = \mathbb{E}\left[\boldsymbol{p}_0^\top \min\left(B, \log\frac{\boldsymbol{p}_0}{\hat{\boldsymbol{p}}}\right)\right] \tag{4}$$

Our main theorem establishes that, with regularity assumptions on the true reward function, an MLP reward model can accurately predict preference probabilities, as measured by truncated KL risk.

**Theorem 2.2** (Truncated KL risk bound, informal). *Suppose the true utility function induced probability of preference satisfies smoothness and regularity assumptions detailed in Assumption C.5 with smoothness characterised by constant $\beta$ and regularity characterised by constant $\alpha$ with dimension of embedding being $d$. Let $\hat{p}$ be an estimator from the family of MLP satisfying regularity assumptions detailed in Assumption C.2 with depth $L$. Define $\phi_n := 2^{\frac{(1+\alpha)\beta + (3+\alpha)d}{(1+\alpha)\beta + d}} n^{-\frac{(1+\alpha)\beta}{(1+\alpha)\beta + d}}$ For sufficiently large $n$, there exists constants $C', C''$ such that when $\Delta_n(\hat{p}, p_0) \leq C'' B \phi_n L \log^2(n)$ then*

$$R_B(\boldsymbol{p}_0, \hat{\boldsymbol{p}}) \leq C' B \phi_n L \log^2(n) \tag{5}$$

*where $a \lesssim b$ means there exists some constant $C$ s.t. $a \leq Cb$ and $a \asymp b$ means $a \lesssim b$ and $b \lesssim a$.*

A detailed formal statement and proof are given as Theorem C.11 in the appendix.

**Corollary 2.3** (Connecting probability to reward). *Given that $(\sqrt{a} - \sqrt{b})^2 = (a - b)^2/(\sqrt{a} + \sqrt{b})^2$, we can apply Lemma C.12 to demonstrate that in a large subset of the embedding space,*

$$\left|p_0(\boldsymbol{\Psi}_1, \boldsymbol{\Psi}_2) - \hat{p}(\boldsymbol{\Psi}_1, \boldsymbol{\Psi}_2)\right| \lesssim \left|\sqrt{p_0} + \sqrt{\hat{p}}\right| \sqrt{\phi_n L} \log(n) \tag{6}$$

$$\left|r(\boldsymbol{\Psi}_1) - r(\boldsymbol{\Psi}_2) - (\hat{r}(\boldsymbol{\Psi}_1) - \hat{r}(\boldsymbol{\Psi}_2))\right| \lesssim \frac{\left|\sqrt{p_0} + \sqrt{\hat{p}}\right|}{\tilde{p}(1 - \tilde{p})} \sqrt{\phi_n L} \log(n) \tag{7}$$

*where $\tilde{p}$ is a probability between $p_0$ and $\hat{p}$, the second line is due to mean value theorem. This indicates that the comparison should be between those pairs that are relatively close in reward to avoid diverging behavior of the logit function.*

Formal proofs and detailed theoretical analyses are provided in Appendix C.

## 3 RETHINKING REWARD MODELING OBJECTIVES IN LLM ALIGNMENT

Practical implementation of the BT model poses several requirements including the paired data, the specially designed anti-symmetric model structure, and the inherent assumptions of the BT model itself. This leads us to question whether we can have alternative approaches to reward modeling. To address this, it is helpful to pause and reflect on the essential requirements of a reward model in LLM alignment. Our data consists of binary preferences, representing the relative ranking between two prompt-response pairs. We can view this as a form of binary classification, with the ultimate goal being to learn a continuous score for *optimization*. While the Bradley-Terry model serves this purpose through its log-likelihood loss and anti-symmetric structure, we now consider whether these formal elements are indispensable and explore possible simplifications.

### 3.1 THE UNIFIED TARGET OF ORDER CONSISTENCY

In basic binary classification, we prioritize accuracy over modeling output probabilities precisely. For example, neural classifiers, despite being overconfident (Guo et al., 2017), are widely used for their accuracy. Similarly, **we don't require the BT model to predict comparison probabilities accurately, but rather to provide a reliable signal for ranking LLM outputs at inference.**

Since our goal is response optimization using a reward proxy, it is sufficient to learn the reward function up to a monotonic transformation. While this might alter preference probabilities, it won't affect optimization results. To this end, the learned reward function $\hat{r}$ only needs to satisfy the following condition: for any two distinct prompt-response pairs $(x_1, y_1)$ and $(x_2, y_2)$, we require that $(\hat{r}(x_1, y_1) - \hat{r}(x_2, y_2))(r(x_1, y_1) - r(x_2, y_2)) > 0$. In other words, the learned reward function must preserve the ordering as the true reward function.

This condition implies the existence of a strictly monotonic increasing function $h$ such that $\hat{r}(\cdot) = h(r(\cdot))$. Such an equivalence is sufficient for optimizing the reward in settings such as sampling-based optimization, and contextual bandits (Agarwal et al., 2014; Lattimore & Szepesvári, 2020). Ideally, if we have access to the ground truth ordering, we can define $h = \text{sign}(r(x_1, y_1) - r(x_2, y_2))$. If we can (1) construct a model $\hat{H} : \mathcal{X} \times \mathcal{Y} \times \mathcal{X} \times \mathcal{Y} \mapsto \{+1, -1\}$ that predicts the correct ordering with high accuracy (i.e., $\hat{H}$ is *order consistent*), and (2) map this ordering into a continuous value, then we can meet the requirements for downstream optimization.

However, this observed ordering is often subject to noise from human annotators. Drawing on insights from the psychological bottleneck literature (Stewart et al., 2005; Guest et al., 2016), it is reasonable to assume that when the true underlying scores of two responses are similar, it becomes more difficult for annotators to distinguish between them. Formally, we have

**Assumption 3.1** (Imperfect Preference Annotation in Approximating True Scores). Denote the true utility difference $\Delta r := |r(x_1, y_1) - r(x_2, y_2)|$, and the annotator function $h(x_1, x_2, y_1, y_2)$ provides feedback that probabilistically aligns with the oracle utility $r(x, y)$. We will assume it is harder for them to assign correctly when the reward difference between two pairs is $\Delta r$ according to:

$$\mathbb{P}\left(h(x_1, x_2, y_1, y_2)(r(x_1, y_1) - r(x_2, y_2)) > 0 \Big| \Delta r\right) = \xi(\Delta r), \tag{8}$$

where $r$ is the unknown oracle utility function, and $\xi(\cdot)$ any monotonic increasing function to $[0.5, 1]$.

With those noisy annotations, the best we can consider is an order consistent with the noisy ordering:

**Definition 3.2** (Order Consistency). We consider the loss over an ordering model $\hat{H}$

$$\mathcal{L}_{\text{oc}}(\hat{r}) = \mathbb{E}_{x_1, x_2, y_1, y_2, h} \mathbb{1}\left[h = \hat{H}\right] \tag{9}$$

That is, the probability that a reward model ordering agrees with annotation.

We show with the following proposition that minimizing this loss would help us achieve order consistency with true reward function:

**Proposition 3.3** (Lower bound on population level order consistency ). *Suppose a learned model $\hat{H}$ achieves objective equation 9 up to $1 - \delta\epsilon$ error for some small $0 < \delta < 1$ and $\epsilon < 3/20$, i.e.,*

$$\mathbb{E}_{x_1, x_2, y_1, y_2, h} \mathbb{1}\left[h = \hat{H}\right] \geq 1 - \delta\epsilon \tag{10}$$

*Then, with probability at least $1 - \delta$ over $\Delta r$, for any given $\Delta r$ the order consistency of $\hat{r}$ with respect to the oracle utility is bounded below by:*

$$\mathbb{E}_{x_1,x_2,y_1,y_2} \left[ \mathbb{1} \left( \hat{H} \cdot [r(x_1,y_1) - r(x_2,y_2)] \geq 0 \right) \bigg| \Delta r \right] \geq (1 - \epsilon) \cdot \xi^2(\Delta r) + \epsilon \cdot (1 - \xi(\Delta r))^2 \quad (11)$$

*Further if we assume that with probability at least $1 - \kappa$, that $\xi(\Delta r) \geq \sqrt{\epsilon^2 + 1 - 3\epsilon} + \epsilon$, we have*

$$\mathbb{E}_{x_1,x_2,y_1,y_2 \sim \ell(x)} \left[ \mathbb{1} \left( \hat{H} \cdot [r(x_1,y_1) - r(x_2,y_2)] > 0 \right) \right] \geq 1 - 4\epsilon - \kappa - \delta \quad (12)$$

In classical classification problems, accuracy is often optimized indirectly using losses like cross-entropy. This hints us to apply techniques for improving prediction accuracy to improve reward modeling. While the BT model uses cross-entropy loss and an antisymmetric structure, in the following, we show an alternative choice that could lead to a simple classification-based algorithm.

### 3.2 THE BT MODEL AS A CHOICE

The BT model is designed to enforce order consistency in the following way: it models the probability that $h = 1$ using $\sigma(\hat{r}_{\text{BT}}(x_1,y_1) - \hat{r}_{\text{BT}}(x_2,y_2))$, where $\sigma$ is the sigmoid function. This allows training with a binary cross-entropy loss:

$$\mathcal{L}_{\text{BT}} = \mathbb{E} \left[ \mathbb{1}_{h=1} \sigma(\hat{r}_{\text{BT}}(x_1,y_1) - \hat{r}_{\text{BT}}(x_2,y_2)) + \mathbb{1}_{h=-1}(1 - \sigma(\hat{r}_{\text{BT}}(x_1,y_1) - \hat{r}_{\text{BT}}(x_2,y_2))) \right] \quad (13)$$

This structure guarantees that flipping the comparison order will also flip the prediction.

### 3.3 RELAXING THE ANTI-SYMMETRY CONSTRAINT: A CLASSIFICATION-BASED METHOD

BT's difference-in-reward structure inherently enforces antisymmetry. To better understand this, consider an order model $\hat{H}$ that outputs a preference vector for both prompt-response pairs, i.e., $\hat{H} := (\hat{H}_1, \hat{H}_2)$, where $\hat{H}_1, \hat{H}_2 : \mathcal{X} \times \mathcal{Y} \mapsto \{1, -1\}$ and ideally align with $(h, -h)$. The BT model imposes a hard constraint such that $\hat{H}_1 = -\hat{H}_2$. With sufficient data, instead of explicitly enforcing this constraint, the structure could be learned implicitly by ensuring order consistency, i.e., $\hat{H}_1 \approx h$ and $\hat{H}_2 \approx -h$. Consider a single model e.g., a neural network or tree $\hat{H}_{\text{clf}}$. Under this construction, the order consistency could be written as $\mathcal{L}_{\text{oc}} := \mathbb{E}(h = \hat{H}_{\text{clf}}(x_1,y_1) \wedge -h = \hat{H}_{\text{clf}}(x_2,y_2))$, we could train a model targeting on predicting these two parts separately, and if the model predicts well, it should satisfy the antisymmetry constraint. A union bound of order consistency is

$$\mathcal{L}_{\text{oc}} \leq \mathcal{L}_{\text{clf}} := \mathbb{E}(h = \hat{H}_{\text{clf}}(x_1,y_1)) + \mathbb{E}(-h = \hat{H}_{\text{clf}}(x_2,y_2)) \quad (14)$$

instead of directly enforcing order consistency, we can use the classification accuracy of each prompt-response pair as a surrogate. In practice, this means training a classifier by treating the annotations and prompt-response pairs independently. Then the logit can be used as a proxy for the reward model. For an alternative perspective: instead of learning the joint probability $\mathbb{P}(i \succ j)$ that depends on both players $i$ and $j$, we focus on learning the marginal probability $\mathbb{P}(i \text{ wins})$. These two are related via Jensen's inequality, with further details provided in Proposition D.1.

In both BT and classification-based reward modeling, there is no theoretical requirement to limit comparisons to the same prompts. For classification models, this is intuitive, as they do not rely on paired data. Similarly, in traditional BT applications, random pairwise comparisons among players are common. This further motivates our investigation into how random comparisons across different prompts affect reward modeling performance.

## 4 EXPERIMENTS

**Objectives of Experiments** In this section, we present empirical results to validate our key insights and methodological contributions. We begin by elaborating on our high-level design motivations for experiments. We aim to address the following questions:

1. How effective are different learning objectives given by the order consistency framework? Specifically, how does the performance of a classification-based model compare to the widely used BT model in reward modeling? (Section 4.1)
2. How do various reward modeling methods perform as annotation quality and availability changes? (Section 4.2)
3. How good is the cross-prompt comparison in annotations? What is essential in determining whether this annotation design is beneficial or not in practice when it is available? (Section 4.3)

**Key Design Principles: Reproducibility, Controllability, and Computational Efficiency**    Our experiments are designed with three key desiderata in mind: high reproducibility, controllability to enable diverse ablation studies and computational efficiency. We prioritized these principles in all aspects of our experimental setup:

- **Base Models.**    We conducted experiments using three open-source LLMs at different scales: Gemma2b, Gemma7b, and LLaMA3-8b (Team et al., 2024; Meta, 2024), alongside their SFT-ed versions (Gemma2b-SFT, Gemma7b-SFT, LLaMA3-8b-SFT), following the setup in Stiennon et al. (2020). Details are provided in Appendix E.
- **Annotations.** To ensure controllability in generation and annotation, and closely simulate human annotation processes, we followed approaches from Gao et al. (2023); Liu et al. (2023); Tran & Chris Glaze (2024); Dong et al. (2024) to utilize open-source golden reward models as annotators to maintain affordability for the community.
- **Datasets.**    We used the `Anthropic-Harmless` and `Anthropic-Helpful` datasets (Bai et al., 2022a), as these are extensively studied in the context of reward modeling, and open-source golden reward models are available (Yang et al., 2024b; Dong et al., 2023; 2024).
- **Evaluation of Learned Reward Models.**    We evaluated the effectiveness of different reward models using Best-of-N (BoN) sampling following Gao et al. (2023). This choice was driven by three key considerations:
  - (1) Performance: Empirical studies show BoN achieves better performance than PPO (Dong et al., 2023; Yuan et al., 2023; Gao et al., 2023; Coste et al., 2023).
  - (2) Stability and Reduced Engineering Overhead: BoN requires no hyperparameter tuning and is more stable than PPO, leading to more consistent and interpretable results (Ivison et al., 2024; Xu et al., 2024).
  - (3) Computational Efficiency and Reproducibility: BoN's reusability across N generations during test time makes it more computationally efficient compared to policy-gradient optimizations (Li et al., 2023; Stiennon et al., 2020). In contrast, using PPO (Schulman et al., 2017) for our 12,000 experimental setups would be computationally prohibitive since each setup requires distinct LLM fine-tuning.

For more details on experiment setup, please refer to Appendix E.

### 4.1    COMPARING THE BRADLEY-TERRY AND CLASSIFICATION OBJECTIVES

**Experiment Setups.**    In this section, we compare reward models trained with different learning objectives — the BT model and the Classification model — as well as their different implementations. For the BT model, the Siamese structure (Bromley et al., 1993) is required, making MLP the only viable backbone implementation (denoted in the plots as **BT-MLP**). For a fair and clear comparison to BT and to isolate the source of gains from implementation, we implement the classification model with both MLP (denoted in the plots as **CLF-MLP**) and the LightGBM (Ke et al., 2017) (denoted in the plots as **CLF-LGB**) given its wide success in machine learning applications (Bentéjac et al., 2021) and especially its successful application in embedding-based reward modeling (Sun et al., 2023). We evaluate those reward models using BoN with N=500, reporting improvements over base models to provide a clear comparison of their performance.

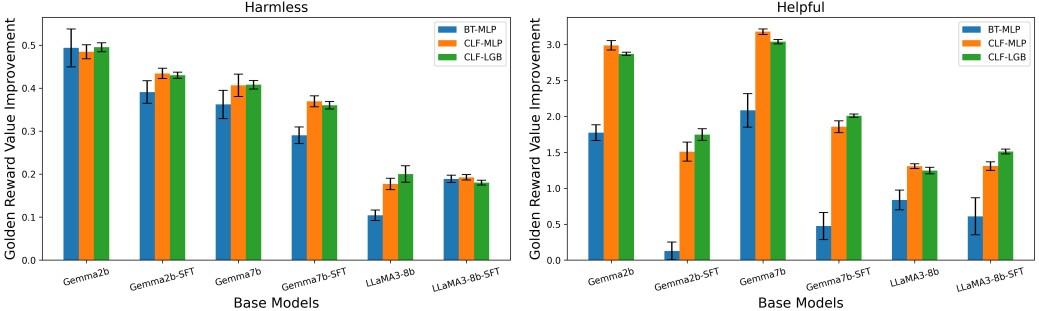

Figure 1: *Comparison between BT and Classification (CLF) reward models.* In general, the classification reward models achieve better performance than the BT reward models, with the added flexibility of using off-the-shelf classifiers beyond MLPs. Error bars are given by 5 runs with different seeds.

**Results and Takeaways.** Figure 1 presents the results on both datasets. The x-axis shows different base models, and the y-axis shows the **improvement on golden reward values** on test prompts using BoN (i.e., the relative performance gains achieved through the reward models). The results indicate that classification-based reward models not only perform better than the BT reward models but also offer greater flexibility by allowing the use of diverse off-the-shelf machine learning algorithms. This makes them a competent alternative to BT models in reward modeling.

## 4.2 HOW ANNOTATION QUALITIES AND QUANTITIES AFFECT PERFORMANCE

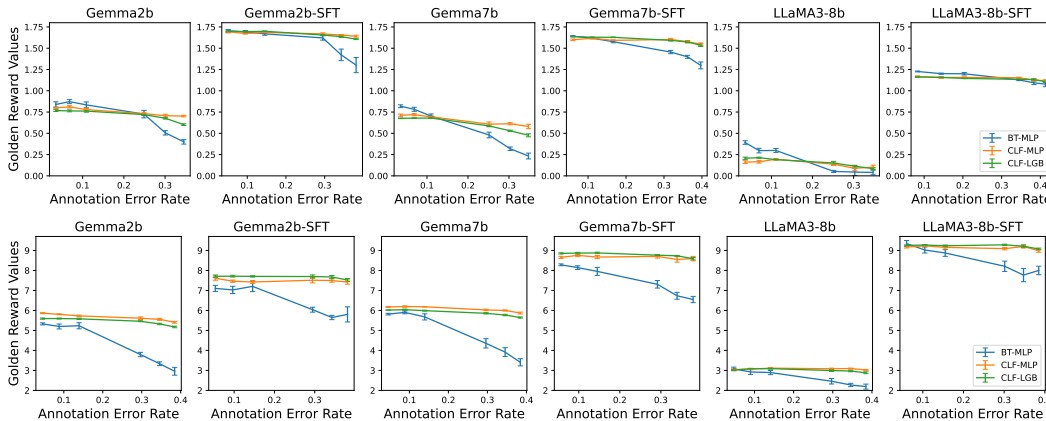

Figure 2: *Changing the annotation quality.* Top row: Harmless; Bottom row: Helpful. Results with 5 seeds.

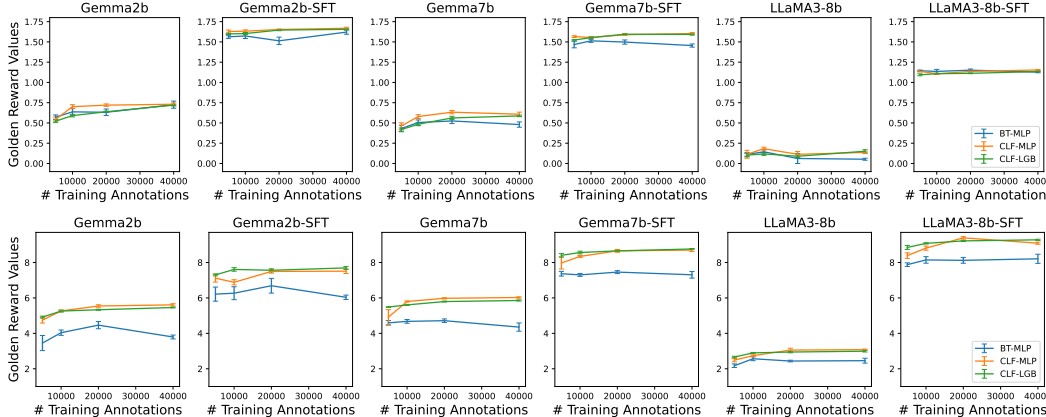

Figure 3: *Changing annotation availability.* Top row: Harmless; Bottom row: Helpful. Results with 5 seeds.

**Experiment Setups.** In this section, we systematically vary both the quality and quantity of annotations to empirically assess the performance of different reward models under diverse conditions. For annotation quality, we use the sigmoid instantiation of equation 8, i.e., $\xi(\Delta r) = \sigma(\beta \Delta r)$, and vary the annotation quality parameter $\beta$ over the range $[0.5, 0.7, 1.0, 3.0, 5.0, 10.0]$, using a fixed set of 40000 annotations for training. These $\beta$ values correspond to annotation error rates ranging from 5% to 38%, which aligns with realistic human annotations (Zheng et al., 2023; Coste et al., 2023; Dubois et al., 2024). To explore the impact of annotation quantity, we experiment with datasets containing $[5000, 10000, 20000, 40000]$ annotations, while holding $\beta = 1$ constant. Additional results cross-sweeping those two set-ups can be found in Appendix F.

**Results and Takeaways.** Figure 2 presents the results of varying annotation quality. From the plots, it is evident that as annotation error rates increase, the classification models exhibit greater robustness compared to the BT models, experiencing smaller performance drops. On the other hand, the BT models outperform classification models when annotation quality is high (i.e., less than 10% wrong labels). Figure 3 shows the results of varying annotation quantities. We can conclude from the results that the classification models consistently outperform BT models, not only delivering superior performance with the same number of annotations but also demonstrating more consistent improvements as the number of annotations increases.

### 4.3 LLM ALIGNMENT WITH PREFERENCE ANNOTATIONS BETWEEN DIFFERENT PROMPTS

**Experiment Setups.** In this section, we examine how cross-prompt comparisons, as opposed to annotations on the same prompt, affect the performance of different reward models. Specifically, for each training prompt, two responses are randomly generated by the LLMs and then presented to annotators (the golden reward model) for preference labeling. In the standard reward modeling setup, the annotators label response pairs generated from the same prompt. In the cross-prompt annotation setup (denoted by the postfix **X-Prompt** in the legends), we randomly select two prompt-response pairs from the dataset for comparison and annotation. We present the results with $\beta = 1$ in the preference annotation processes and $40000$ pairs of annotations in our main text as the most common setting. We provide full results over other experimental configurations in Appendix F.

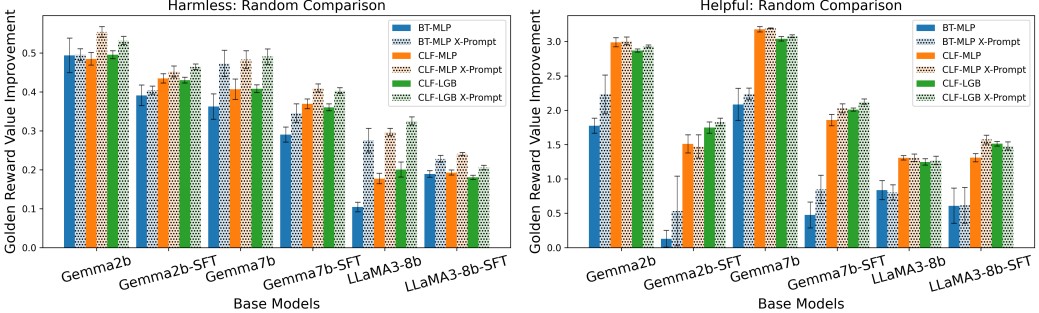

Figure 4: *Results comparing cross-prompt comparison based annotations.* Preference annotations on cross-prompt comparisons outperform same-prompt comparisons. Error bars are from 5 runs with different seeds.

**Results and Takeaways.** Figure 4 shows the results across 2 datasets, 6 base models, and 3 different reward modeling methods. Shaded plots represent results from cross-prompt comparisons. The y-axis measures golden reward value improvements achieved through BoN sampling using the respective reward models. From these results, it is clear that cross-prompt comparisons outperform same-prompt comparisons, offering substantial improvements in reward model performance.

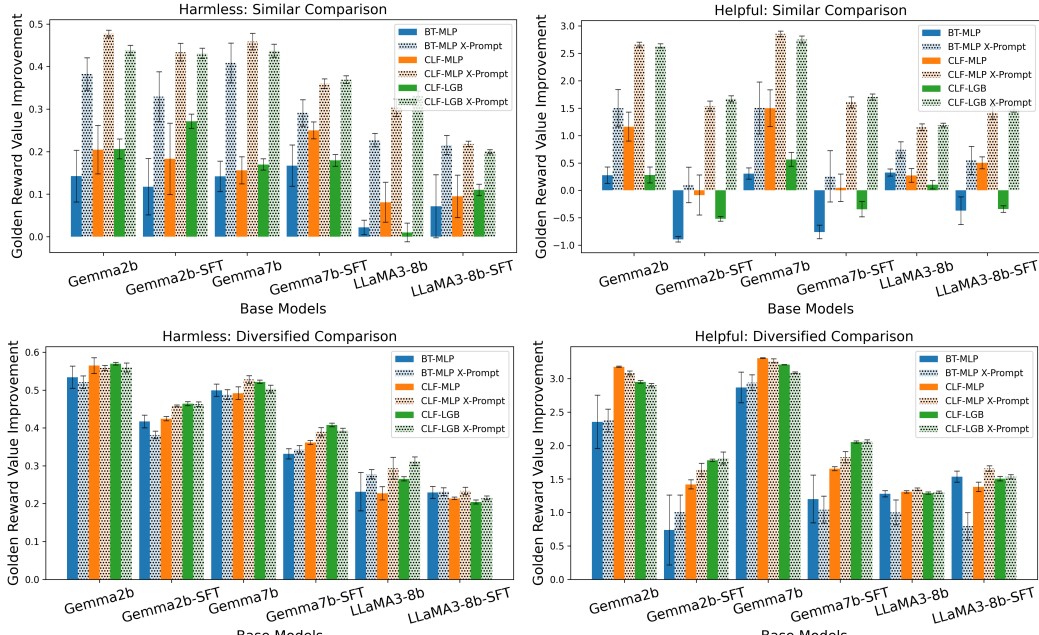

Figure 5: *Results comparing cross-prompt comparison-based annotations on synthetically generated similar or diversified comparison pairs.* Cross-prompt comparison significantly improves the performance of reward modeling with same-prompt response pairs lacking diversity. Error bars are from 5 runs with different seeds.

**Further Investigation.** Intuitively, the cross-prompt comparison can improve the quality of annotation since it increases the diversity in generation. To further explore this, we introduce two *synthetic*

*setups*[2] designed to analyze the source of gains. In the **Similar Comparison** setup, response pairs generated for a single prompt exhibit similar quality and lack diversity. In the **Diversified Comparison** setup, response pairs for a single prompt are of different qualities. Specifically, we use a golden reward model to score 10 generated responses per prompt and select the two middle-ranked responses to simulate the **Similar Comparison** setup. Whereas in the **Diversified Comparison** setting, we use the highest and lowest-scored responses to construct the response pair for each prompt.

**Results and Takeaways.** Figure 5 shows the results of those two synthetic setups. We can conclude that cross-prompt comparisons are essential when responses for a single prompt lack diversity. In the **Similar Comparison** setting, the conventional same-prompt comparisons often fail to produce informative reward models that can improve golden reward values. In contrast, cross-prompt annotations substantially enhance performance in such cases. In the **Diversified Comparison** setting, the need for cross-prompt comparisons diminishes as response diversity increases, though they do not have an evident negative impact on reward modeling performance. Comparing results across both settings, as well as results in Figure 4, we find the superiority of cross-prompt comparison is reflected in its general capability regardless of response diversities — the performance achieved by cross-prompt annotations is more stable and has lower dependence of the response diversities.

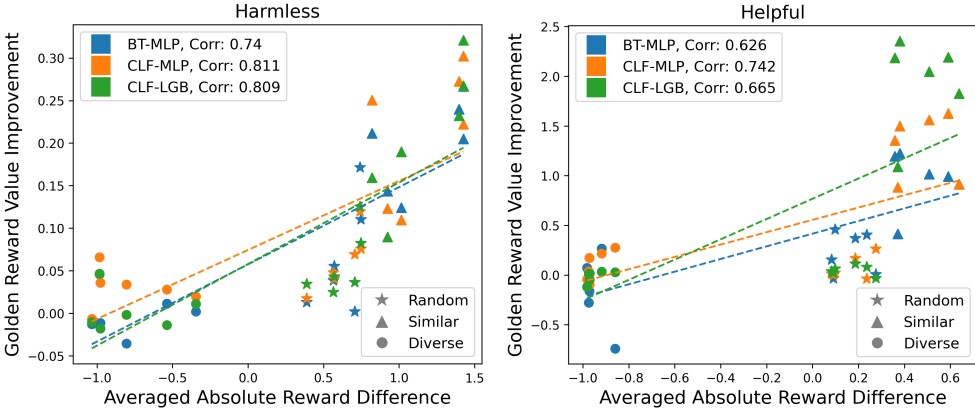

Figure 6: *Comparing the averaged absolute difference in scores in pairwise annotations (x-axis) and improvements achieved by using cross-prompt annotations (y-axis).* The two variables are highly correlated.

Finally, we examine the relationship between the *average absolute score differences in pairwise annotations* (x-axis) across three setups (i.e., **Random** for randomly select two responses for each pair, **Similar** and **Diverse** for the two synthetic settings) and the corresponding *performance improvements from cross-prompt annotations* (y-axis). Scatter plots with linear fits are shown in Figure 6. The strong correlation indicates that cross-prompt annotations are most beneficial when same-prompt responses lack diversity. Importantly, we find the averaged reward differences between pairwise data in the synthetic setting of **Similar** cases and the **Random** settings are similar, this implies that in practice, when randomly selecting two responses for a single prompt to be annotated, those pairs are likely facing the challenge of high response similarity. And this is the case when cross-prompt annotation can be applied to improve performance.

**Conclusive Remark on Experiments** Our experimental results highlight the advantages of classification-based reward models over traditional BT models, particularly in terms of flexibility and robustness to annotation quality and quantity. While BT models perform better under high-quality annotations, classification models demonstrate superior overall performance and resilience to increasing annotation error rates. Additionally, our empirical studies on cross-prompt comparisons show it significantly improves reward modeling, especially when responses to the same prompt lack diversity. Through synthetic experiments, we further demonstrate that the challenge of limited diversity is likely to occur in practice, providing additional justification for exploring this annotation method in future research.

---

[2]It is worth noting that none of those two synthetic setups is realizable in practice without a reward model. And the randomly sampled pairs are the only realistic setups for annotations.

ACKNOWLEDGMENT

We sincerely appreciate the efforts of our anonymous reviewers, ACs, and SACs for their valuable feedback and constructive suggestions, which have made our paper more complete. We thank Ziping Xu and Jalaj Bhandari for their early-stage discussions and encouragement during the 1st RL Conference, which was crucial in motivating this paper. Additionally, we are grateful to Rui Yang and Ruocheng Guo for their insightful discussions that contributed to the development of this research.

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

## A  RELATED WORK

**The Bradley-Terry Model in RLHF**  Hark back to the seminal paper on RLHF (Christiano et al., 2017), the motivation for using the Bradley-Terry-*type* model is to understand it as a specialization of the Luce-Shephard choice rule (Luce, 1959; Shepard, 1957). In a simplified version of the two-option setting, the Luce-Shephard choice rule says the probability of choosing a particular option from a set of alternatives is proportional to the utility assigned to that option. The application of such a rule is well aligned with the Bradley-Terry model Bradley & Terry (1952) where pairwise comparisons between players can be used to estimate their ability (utility) scores, where a game is inherently stochastic while the ability of a player is fixed. In later work of alignment from human feedback Stiennon et al. (2020); Ouyang et al. (2022); Rafailov et al. (2024), such a model is directly applied and has achieved great success in improving the quality of LLMs in various tasks when direct evaluation is impossible.

The usage of the Bradley-Terry model in RLHF has been challenged in the literature from different perspectives: Munos et al. (2023) points that the Bradley-Terry model can not capture non-transitive preferences, and maximizing the corresponding Elo score can be a different objective as compared to the objective of optimizing preferences. Azar et al. (2023) points out that using the Bradley-Terry modelization in the direct preference optimization methods (Rafailov et al., 2024) could lead to problematic overfitting when sampled preferences are deterministic.

**Bradley-Terry Model for Parameter Estimation and Prediction**  The most classic Bradley-Terry model suitable for arena setting has been extensively studied theoretically started by Bradley & Terry (1952) themselves. Ford Jr (1957) established identifiability and asymptotic theory. More recently Wu et al. (2022) compared asymptotic theory under different identifiability assumptions. Simons & Yao (1999) studied Bradley-Terry when number of players diverge under the assumption number of competitions each players played also diverge. In contrast Han et al. (2020) studied the setting when comparisons are sparse and proposed a consistent procedure needs only $O(N\log^3(N))$ comparisons. On prediction side, using features to predict ability was explored soon after Bradley & Terry (1952), Springall (1973) assumed ability was a linear combination of features while De Soete & Winsberg (1993) used spline functions. Bockenholt (1988) cast Bradley-Terry as a special logistic regression problem and opened investigation from this len. Chen & Pouzo (2012) developed theory for a general class of nonparametric models including logistic regression. On deep learning side Schmidt-Hieber (2020) studied asymptotic theory on deep neural network for nonparametric regression and a follow up paper Bos & Schmidt-Hieber (2022) studied nonparametric logistic regression using deep neural net. However these theory cannot be directly applied Bradley-Terry model because we need a unique reward model for all prompt-response pairs and the probability is not arbitrary but soft max from passing two pairs to a *single* network. Studies like Bos & Schmidt-Hieber (2022) could be seen as two pairs passing two arbitrary neural networks rather than one.

**Non-Pairwise Data and Cross-Prompt Comparisons in RLHF**  Non-pairwise data (Ethayarajh et al., 2024) and cross-prompt comparisons (Yin et al., 2024) were explored in the literature. The KTO (Ethayarajh et al., 2024) is rooted in prospect theory (Tversky & Kahneman, 1992) that risk-averse human prospective theories assign different values to losses and gains. And RPO (Yin et al., 2024) proposed to reflect the complex nature of human learning in alignment through involving response comparisons among both identical and similar questions. On one hand, RPO supports our insight into making cross-prompt comparisons in practice by drawing inspiration from human learning. On the other hand, the motivations and implementations between our work and RPO are both different: in our work, the cross-prompt annotation practice is motivated by a *"why not"* observation — given the fact that there is no specialty of considering single-prompt comparisons in our order-consistency theory and the BT model for predictions. Besides different motivations, there are several significant differences between RPO and our cross-prompt comparison: RPO does not study the annotation efficiency, or what is the optimal way of doing annotation under a budget, yet our focus is on studying the cross-prompt annotation as an alternative way of doing annotations. Moreover, RPO considers a strategic re-weighting method based on embedding space similarity among responses, leading to a direct alignment method. All of those works are direct alignment methods (Zheng et al., 2024; Liu et al., 2024; Zhao et al., 2023; Azar et al., 2023; Ethayarajh et al., 2024), which do not explicitly build reward models as their intermediate steps. One potential drawback of those methods without explicit reward models is that it will be challenging or impossible to

conduct inference-time optimization (Liu et al., 2023) — especially when compared with our embedding based light-weighted reward models. And recent work has demonstrated the superiority of two stage methods over direct alignment methods (Ivison et al., 2024).

**Representation Learning in Reward Modeling**    In our work, we separate the reward model learning from the representation learning task and focus only on the reward modeling part to better isolate the source of gains. Recent advance in generative reward models highlight it is possible to use generation tasks to regularize the learned embeddings and improve reward modeling performance (Yang et al., 2024a). Also in Zhang et al. (2024), the generative verifiers construct reward predictions through next-token prediction to maximally leverage the ability of LLM as token generators to improve their evaluation ability. While our research on the reward models is orthogonal to this line of research, future exploration on combining either embedding learning or generative ability of LLMs with different order consistency objectives could be highly promising directions to enjoy the improvements from both sides.

# B    UNDERLYING ASSUMPTIONS OF APPLYING THE BRADLEY-TERRY MODEL IN PREFERENCE ANNOTATIONS

Different from those prior works that challenge the *practical application* of the BT model Azar et al. (2023); Munos et al. (2023); Tang et al. (2024), we revisit the basic assumptions of modeling human preference annotations with the BT model, and answer the following questions:

*What are the underlying assumptions when we assume the BT model can be used to model the preference annotations?*

The canonical interpretation of how to apply equation 1 in preference-based learning is that: when randomly sampling a human annotator from the population, the human annotator's choice of the preferred response is proportional to the response's utility value. Formally, we use $x, y_1, y_2$ to denote the prompt and responses, the above interpretation implies the following assumptions:

**Assumption B.1** (Existence of a Deterministic Oracle Utility Value). Given $x$, the utility value of any response $y$ exists and is deterministic. i.e., $r_{x,y}, \forall x, y$ is a deterministic real value.

**Definition B.2** (Individual Annotation under Bias). For annotator $A$, the annotation result is deterministic and depends on the comparison of their biased evaluation of the utility values of both responses $y_1$ and $y_2$, according to

$$P_A(y_1 \succ y_2 | x) = \mathbb{1}(r_{x,y_1}^{(A)} > r_{x,y_2}^{(A)}) = \mathbb{1}(r_{x,y_1} - r_{x,y_2} > b(x, y_1, A) - b(x, y_2, A)) \tag{15}$$

Based on the above definitions, we can now explicitly write the assumption of applying the BT model to this randomized annotation process:

**Assumption B.3** (Logistic Difference Assumption). The $b(x, y_1, A) - b(x, y_2, A)$ is sampled iid from a standard logistic distribution for all $x, y, A$:

$$P(b(x, y_1, A) - b(x, y_2, A) \le t) = \frac{1}{1 + e^{-t}} \tag{16}$$

**Remark B.4.** (Transitive property of difference) A reader might be (rightfully) worried if this assumption is consistent with transitive property of difference, e.g., when considering multiple comparisons we have to have $b(x, y_1, A) - b(x, y_3, A) = b(x, y_1, A) - b(x, y_2, A) + b(x, y_2, A) - b(x, y_3, A)$ while knowing sum of (independent) logistic distributions is not logistic. One should be aware that the two terms being summed are not independent and the assumption can be achieved by assuming all annotator biases are independently Gumbel distributed with the same scale parameter.

With this assumption, we arrive at the BT-*type* model

**Proposition B.5** (BT-*type* Model on Annotation under Logistic Difference Assumption).

$$P(y_1 \succ y_2 | x) = P(r_{x,y_1} - r_{x,y_2} > b(x, y_1, A) - b(x, y_2, A)) = \frac{1}{1 + e^{-(r_{x,y_1} - r_{x,y_2})}} \tag{17}$$

In comparison, we can alternatively have the following assumption and have another model:

**Assumption B.6** (Gaussian Difference Assumption). The $b(x, y_1, A) - b(x, y_2, A)$ is sampled from a standard Gaussian distribution:

$$b(x, y_1, A) - b(x, y_2, A) \sim \mathcal{N}(0, 1) \tag{18}$$

**Proposition B.7** (Model on Annotation under Gaussian Difference Assumption).

$$P(y_1 \succ y_2 | x) = P\big(r_{x,y_1} - r_{x,y_2} > b(x, y_1, A) - b(x, y_2, A)\big) = \Phi(r_{x,y_1} - r_{x,y_2}), \tag{19}$$

*where $\Phi$ is the CDF of the Gaussian distribution.*

The next paragraph elaborates implications of those different assumptions.

**Understanding the BT Model from a Gaussian Assumption on Players' Performance in a Game**  Technically, the BT model assumes that a player's ability **in a specific game** can be represented by a score. To better understand the rationale behind the BT model, we can take a step back and consider a Gaussian assumption for game performances. Specifically, we assume that in each two-player game, a player's performance can be modeled as a Gaussian distribution centered on their score, with the variance of these distributions capturing the stochastic nature of the game and the variability in the players' performances.

For instance, when player $A$ with a score of $S_A$ and variance $\sigma_A$ competes against player $B$ with a score of $S_B$ and variance $\sigma_B$, the probability that $A$ defeats $B$ in a game (denoted as $A \succ B$), given the Gaussian assumption on performance, is

$$P(A \succ B) = P\left(x_a \geq x_b | x_a \sim N(S_A, \sigma_A^2), x_b \sim N(S_B, \sigma_B^2)\right) = \frac{1}{2} + \frac{1}{2}\mathrm{erf}\left(\frac{S_A - S_B}{\sqrt{2(\sigma_A^2 + \sigma_B^2)}}\right). \tag{20}$$

This is known to be the Thurstonian model (Thurstone, 1927). In practice, if we further **assume** that the stochasticity characterized by performance variance is game-dependent rather than player-dependent, the denominator in equation 20 becomes a player-agnostic constant, which can be absorbed into the players' scores. In other words, **the absolute values in the scoring system are scaled by the inherent variability of the game.**

In the BT model, the $\tanh(\cdot)$ rather than the error function $\mathrm{erf}(\cdot)$ is used, for the sake of a better empirical fit and mathematical convenience.[3] Formally, the BT model is

$$P(A \succ B) = \frac{1}{1 + e^{-(S_A - S_B)}} \tag{21}$$

## C  ASYMPTOTIC THEORY ON BT REWARD MODELING

Recall that we have a *known* embedding function $\Psi(\cdot, \cdot) \, \mathcal{X} \times \mathcal{Y} \to [0, 1]^d$ such that there exists an unknown function $g : \mathbb{R}^d \to \mathbb{R}$ and the true utility function $r(y, x) = g(\Psi(y, x))$ for all $y, x$. We assume the embedding to have a range of $[0, 1]$, if the pretrained model do not have this range, we can scale it to be. Then reward modeling reduce to learn the function $g$. Observe that under this formalism there is no need for a comparison to have the same prompt.

### C.1  ANALYZING BT REWARD MODELING AS NON-PARAMETRIC LOGISTIC REGRESSION WITH ANTI-SYMMETRIC STRUCTURE

Denote our reward model as $\hat{r}_\theta$, parameterized by $\theta$, when there is no confusion, we will abbreviate it as $\hat{r}$. The reward difference between two pairs of prompts and responses $(x_1, y_1)$ $(x_2, y_2)$ is then $\hat{r}^\Delta(\Psi(x_1, y_1), \Psi(x_2, y_2)) := \hat{r}(\Psi(x_1, y_1)) - \hat{r}(\Psi(x_2, y_2))$. We could have $x_1 = x_2$, i.e., having matched prompt but it is not necessary. And the predicted probability that $(x_1, y_1)$ better than $(x_2, y_2)$ is then $\sigma(\hat{r}^\Delta(\Psi(x_1, y_1), \Psi(x_2, y_2)))$ with $\sigma$ being sigmoid function. This is the same as having a softmax over two rewards, i.e., $\sigma(\hat{r}^\Delta(\Psi(x_1, y_1), \Psi(x_2, y_2))) =$

---

[3]So equivalently, the BT model assumes the players' performance differences are sampled from a logistic distribution rather than a Gaussian distribution.

softmax($\hat{r}(\Psi(x_1, y_1)), \hat{r}(\Psi(x_2, y_2))$). The second viewpoint is easier in applying theoretical analysis techniques developed in literature (Schmidt-Hieber, 2020; Bos & Schmidt-Hieber, 2022). To that end, we denote the vector of two rewards as $\hat{r}$ and the class probability is then softmax($\hat{r}$) Training such reward model reduce to train a classifier with cross entropy loss whose predicted conditional class probability being softmax($\hat{r}(\Psi(x_1, y_1)), \hat{r}(\Psi(x_2, y_2))$). Our theoretical development generally follow Bos & Schmidt-Hieber (2022). This is a special case of an MLP that could preserve symmetry such that if we exchange the role of $x_1, y_1$ and $x_2, y_2$ we get a negative difference. By showing this particular logit-Siamese architecture could approximate the true class probability, we can deduce that a BT reward model could approximate true reward.

For notational convenience, we denote the embedding of the $i$th pair as $\Psi_1^{(i)}, \Psi_2^{(i)}$ for $i = 1, \ldots, n$, without lose of generality we assume $\Psi^{(i)} \in [0,1]^d$. Denote the true preference probability as $p_0^{(i)}$ and model predicted as $\hat{p}^{(i)} = (\sigma(\hat{r}^\Delta(\Psi_1^{(i)}, \Psi_2^{(i)})), 1 - \sigma(\hat{r}^\Delta(\Psi_1^{(i)}, \Psi_2^{(i)}))) =$ softmax($\hat{r}(\Psi_1^{(i)}, \Psi_2^{(i)})$) denote the preference vector as $h^{(i)}$, equals to $(1, 0)$ if the first response pair is prefered and $(0, 1)$ otherwise. The BT model is reduced to a (pairwise) classification problem such that the likelihood is given by

$$\widetilde{\mathcal{L}}_{\text{CE}}(p) = -\frac{1}{n} \sum_{i=1}^{n} (h^{(i)})^\top \log(p^{(i)}), \quad \hat{p} = \arg\min_{p \in \mathcal{F}_\theta} \widetilde{\mathcal{L}}_{\text{CE}}(p)$$

It is unrealistic to assume we can find an NN that actually attends the global minimum, we denote the difference between the fitted NN and the global minimum as

$$\Delta_n(p_0, \hat{p}) = \mathbb{E}\left[\widetilde{\mathcal{L}}_{\text{CE}}(\hat{p}) - \min_{p \in \mathcal{F}_\theta} \widetilde{\mathcal{L}}_{\text{CE}}(p)\right]$$

We consider truncated KL risk, similar to Bos & Schmidt-Hieber (2022) to overcome the divergence problem of KL risk.

**Definition C.1** (Truncated KL risk (Bos & Schmidt-Hieber, 2022))**.** The $B-$truncated KL risk for a probability estimator $\hat{p}$

$$R_B(p_0, \hat{p}) = \mathbb{E}\left[p_0^\top \min\left(B, \log\frac{p_0}{\hat{p}}\right)\right] \tag{22}$$

We consider MLP with ReLU activations for $\mathcal{F}_\theta$, depends on depth $L$ and width vector $m = (m_0, \ldots, m_L)$ i.e.,

$$\mathcal{F}(L, m) = \{f : \mathbb{R}^{m_0} \to \mathbb{R}, x \to f(x) = W_L \psi_{v_L} W_{L-1} \ldots W_1 \psi_{v_1} W_0 x\}$$

where $\psi_v(x) = \max(x - v, 0)$ is the ReLU activation with bias $v$.

**Assumption C.2** (MLP reward model)**.** We will further assume the network parameters having norm restriction and sparsity, a common assumption in studying MLP for classification problems (Yara & Terada, 2024; Bos & Schmidt-Hieber, 2022). That is, in this work we consider networks from the family

$$\mathcal{F}(L, m, s) := \left\{f : f \in \mathcal{F}(L, m), \max_{j=0,\ldots,L} \max(\|W_j\|_\infty, |v_j|_\infty) \leq 1, \sum_{j=0}^{L}(\|W_j\|_0 + |v_j|_0) \leq s\right\} \tag{23}$$

Another useful function class in theory is the softmax output version of the reward model, i.e., consider

$$\mathcal{F}_\sigma(L, m, s) := \{p(\Psi_1, \Psi_2) : p = \text{softmax}(f(\Psi_1), f(\Psi_2)), f \in \mathcal{F}(L, m, s)\}$$

Next we assume the probability of preference is not too close to 0 or 1, in the form of a small value bound

**Definition C.3** (Small value bound by Bos & Schmidt-Hieber (2022))**.** Let $\alpha \geq 0$ and $\mathcal{H}$ is a function class we say $\mathcal{H}$ is $\alpha-$small value bounded ($\alpha-$SVB) if there exists a constant $C > 0$ s.t. for all probability estimated $p = (p_0, p_1) \in \mathcal{H}$ it holds that

$$\mathbb{P}(p_k(\Psi_1, \Psi_2) \leq t) \leq Ct^\alpha, \quad \text{for all } t \in (0, 1] \text{ and all } k = 0, 1 \tag{24}$$

which indices that our reward function and design of comparisons should have a tail behavior that we do not tend to compare pairs having very different reward function. We denote this family as

$$\mathcal{S}(\alpha, C) := \left\{ g : \mathbb{P}_{\boldsymbol{\Psi}_1, \boldsymbol{\Psi}_2} \left[ g(\boldsymbol{\Psi}_1) - g(\boldsymbol{\Psi}_2) \leq \sigma^{-1}(t) \right] \leq C t^\alpha \text{ for all } t \in (0, 1] \right\} \tag{25}$$

**Definition C.4** (Hölder smooth function). For $\beta > 0$ and $D \subset \mathbb{R}^d$, the ball of $\beta-$Hölder functions with radius $Q$ is defined as

$$C^\beta(D, Q) := \left\{ f : D \to \mathbb{R} : \sum_{\boldsymbol{\gamma} : ||\boldsymbol{\gamma}||_1 < \beta} ||\partial^\gamma f||_\infty + \sum_{\boldsymbol{\gamma} : ||\boldsymbol{\gamma}||_1 = \lfloor \beta \rfloor} \sup_{\boldsymbol{x} \neq \boldsymbol{y} \in D} \frac{|\partial^\gamma f(\boldsymbol{x}) - \partial^\gamma f(\boldsymbol{y})|}{||\boldsymbol{x} - \boldsymbol{y}||_\infty^{\beta - \lfloor \beta \rfloor}} \leq Q \right\} \tag{26}$$

**Assumption C.5** (Class of true utility function). We assume that the true reward functions are $\beta-$Hölder and the induced probability by softmax is $\alpha-$SVB. I.e., we consider the function class

$$\mathcal{G}_\alpha(\beta, Q, C) = C^\beta(D, Q) \cap \mathcal{S}(\alpha, C) \tag{27}$$

Note that this is nonempty since constant $g$ satisfy the requirement for any $\beta > 0, \alpha > 0$ with $Q > 1/2$ and $C \geq 2^\alpha$.

**Theorem C.6** (Theorem 5 of Schmidt-Hieber (2020), approximating Hölder smooth functions). *For every function $f \in C^\beta(D, Q)$ and every $M > \max((\beta + 1)^\beta, (Q + 1)^{\beta/d} e^\beta)$ there exist a neural network $H \in \mathcal{F}(L, \boldsymbol{m}, s)$ with $L = 3\lceil \log_2(M)(d/\beta + 1)(1 + \lceil \log_2(\max(d, \beta)) \rceil)$, $\boldsymbol{m} = (d, 6(d + \lceil \beta \rceil) \lfloor M^{d/\beta} \rfloor, \ldots, 1)$ and $s \leq 423(d + \beta + 1)^{3+d} M^{d/\beta} \log_2(M)(d/\beta + 1)$ such that*

$$||H - f||_\infty \leq \frac{C_{Q,\beta,d}}{M}$$

**Remark C.7.** Note that since $\mathsf{softmax}$ with two output is Lipchetz with constant 2, we have $L_\infty$ distance between the softmax output being bounded by $2C_{Q,\beta,d}/M$ by applying the above theorem.

**Theorem C.8** (Oracle inequality, theorem 3.5 Bos & Schmidt-Hieber (2022)). *Let $\mathcal{F}$ be a class of conditional class probabilities and $\hat{p}$ be any estimator taking value in $\mathcal{F}$, If $B \geq 2$ and $\mathcal{N}_n = \mathcal{N}(\delta, \log(\mathcal{F}), d_\tau(\cdot, \cdot)) \geq 3$ for $\tau = \log(C_n e^{-B}/n)$, then*

$$R_B(p_0, \hat{p}) \leq (1 + \epsilon) \left( \inf_{p \in \mathcal{F}} + \Delta_n(p_0, \hat{p}) + 3\delta \right)$$
$$+ \frac{(1 + \epsilon)^2}{\epsilon} \frac{68 B \log(\mathcal{N}_n) + 272 B + 3 C_n (\log(n/C_n) + B)}{n} \tag{28}$$

*for all $\delta, \epsilon \in (0, 1]$ and $0 < C_n \leq n/e$.*

**Lemma C.9** (Adapted lemma 3.8 of (Bos & Schmidt-Hieber, 2022)). *Let $V = \prod_{\ell=0}^{L+1}(m_\ell + 1)$, then for every $\delta > 0$*

$$\mathcal{N}(\delta, \log(\mathcal{F}_\sigma(L, \boldsymbol{m}, s)), || \cdot ||_\infty) \leq (8\delta^{-1}(L + 1)V^2)^{s+1} \tag{29}$$

*and*

$$\log \mathcal{N}(\delta, \log(\mathcal{F}_\sigma(L, \boldsymbol{m}, s)), || \cdot ||_\infty) \leq (s + 1) \log(2^{2L+7} \delta^{-1}(L + 1)d^2 s^L) \tag{30}$$

*Substitute to the first bound and take log yield the second line.*

Note that, although the proof largely carries follow Bos & Schmidt-Hieber (2022) we cannot directly apply lemma 3.8 of Bos & Schmidt-Hieber (2022) in our setting because before softmax layer the two one-dimensional scores came from the *same* MLP with identical activation. This is only a subset of the family Bos & Schmidt-Hieber (2022) considered in their work.

*Proof.* For $g \in \log(\mathcal{F}_\sigma(L, \boldsymbol{m}, s))$, there exists an $f_g \in \mathcal{F}(L, \boldsymbol{m}, s)$ such that $g(\boldsymbol{x}_1, \boldsymbol{x}_2) = \log(\mathsf{softmax}(f_g(\boldsymbol{x}_1), f_g(\boldsymbol{x}_2)))$.

By Lemma 5 of Schmidt-Hieber (2020), we have

$$\mathcal{N}(\delta/4, \mathcal{F}(L, \boldsymbol{m}, s), || \cdot ||_\infty) \leq (8\delta^{-1}(L + 1)V^2) \tag{31}$$

Let $\delta > 0$, denote $\{f_j\}_{j=1}^{\mathcal{N}}$ the centroid of a minimum $\delta/4$ cover of $\mathcal{F}(L, \boldsymbol{m}, s)$, thus for each $f_j \in \mathcal{F}(L, \boldsymbol{m}, s)$, there exists a $\hat{f}_j$ s.t., $\hat{f}_j$'s are interior of a $\delta/2$-cover of $\mathcal{F}(L, \boldsymbol{m}, s)$ for a $g \in$

$\log(\mathcal{F}_\sigma(L, \boldsymbol{m}, s))$, by covering property there exists a $j$ s.t. $||f_g - \hat{f}_j||_\infty \leq \delta/2$, by proposition C.6 of Bos & Schmidt-Hieber (2022), we will abbriviate $f_g(x_i)$ as $f_g^i$

$$
\begin{aligned}
||g - \log(\text{softmax}(\hat{f}_j^1, \hat{f}_j^2))||_\infty &= ||\log(\text{softmax}(f_g^1, f_g^2)) - \log(\text{softmax}(\hat{f}_j^1, \hat{f}_j^2))||_\infty \\
&\leq 2||[f_g^1, f_g^2] - [\hat{f}_j^1, \hat{f}_j^2]||_\infty \leq 2\max_i ||f_g^i - \hat{f}_j^i|| \leq \delta
\end{aligned}
\tag{32}
$$

Since $g$ arbitrary, we have $\log(\text{softmax}(\hat{f}_j^1, \hat{f}_j^2))$ is an internal $\delta$ cover of $\log(\mathcal{F}_\sigma(L, \boldsymbol{m}, s))$, hence

$$
\mathcal{N}(\delta, \log(\mathcal{F}_\sigma(L, \boldsymbol{m}, s)), ||\cdot||_\infty) \leq \mathcal{N}(\delta/4, \mathcal{F}(L, \boldsymbol{m}, s), ||\cdot||_\infty) \leq 8\delta^{-1}(L+1)V^2
\tag{33}
$$

The second bound is by having $m_0 = d$, $m_{L+1} = 1$ (since we have scalar reward) and removing inactive nodes we have by proposition A1 of Bos & Schmidt-Hieber (2022) $V \leq ds^L 2^{L+2}$ □

**Remark C.10.** Readers might suspect this is a direct result of Bos & Schmidt-Hieber (2022) by first concatenating two embeddings and training an MLP with this joint embedding to predict softmax score. While this proposal will satisfy the requirements for the theory, it does not provide a way to generate a reward for one embedding that does not depend on another embedding and might not be antisymmetric. Theoretically, the rate depends on embdding dimension $d$ rather than the concatenated dimension $2d$ as directly using results from Bos & Schmidt-Hieber (2022).

**Theorem C.11** (Truncated KL risk bound). *Suppose the true utility function induced probability of preference is in $\mathcal{G}_\alpha(\beta, Q, C)$ for $\alpha \in [0, 1]$ and $\phi_n = 2^{\frac{(1+\alpha)\beta+(3+\alpha)d}{(1+\alpha)\beta+d}} n^{-\frac{(1+\alpha)\beta}{(1+\alpha)\beta+d}}$. Let $\hat{p}$ be an estimator from family $\mathcal{F}_\sigma(L, \boldsymbol{m}, s)$ satisfying 1) $A(d, \beta)\log_2(n) \leq L \lesssim n\phi_n$ for some suitable constant $A(d, \beta)$, 2) $\min_i m_i \gtrsim n\phi_n$ and 3) $s \asymp n\phi_n \log(n)$. For sufficiently large $n$, there exists constants $C', C''$ such that when $\Delta_n(\hat{p}, p_0) \leq C'' B\phi_n L \log^2(n)$ then*

$$
R_B(\boldsymbol{p}_0, \hat{\boldsymbol{p}}) \leq C' B\phi_n L \log^2(n)
\tag{34}
$$

*where $a \lesssim b$ means there exists some constant $C$ s.t. $a \leq Cb$ and $a \asymp b$ means $a \lesssim b$ and $b \lesssim a$.*

*Proof.* We apply oracle bound Theorem C.8. Take $\delta = n^{-1}$ and $\epsilon = C_n = 1$, using the fact that $d_\tau$ is upper bounded by sup-norm. Then apply Lemma C.9, we have

$$
\begin{aligned}
R_B(\boldsymbol{p}, \hat{\boldsymbol{p}}) &\leq 2\left(\inf_{\boldsymbol{p}} R_B(\boldsymbol{p}_0, \boldsymbol{p}) + \Delta_n(\boldsymbol{p}_0, \hat{\boldsymbol{p}} + \frac{3}{n})\right) \\
&\quad + 4 \cdot \frac{68B(s+1)\log(2^{2L+7}\delta^{-1}(L+1)d^2s^L) + 272B + 3(\log(n)+B)}{n}
\end{aligned}
\tag{35}
$$

We pick $M = \lfloor c2^{\frac{(2+\alpha)\beta}{(1+\alpha)\beta+d}} n^{\frac{\beta}{(1+\alpha)\beta+d}} \rfloor$ for small small $c$, with large enough $n$, we apply Theorem C.6, its softmax transformed version denote as $\tilde{p}$ is in $\mathcal{F}_\sigma(L, \boldsymbol{m}, s)$ with $L = $ and maximim width bounded by $\lesssim M^{d/\beta} = c^{d/\beta} n\phi_n$, and similarly we have $s \lesssim c^{d/\beta} M^{d/\beta} \log_2(M) = c^{d/\beta} n\phi_n \log_2(M)$. Whenever we have $A(d, \beta)\log_2(n) \leq L \lesssim n\phi_n$ we have the maximum width is $\gtrsim n\phi_n$ and $s \asymp n\phi_n \log(n)$. Observe that the softmax output network satisfy Theorem 3.2 of Bos & Schmidt-Hieber (2022), we have with $C_1 = 4(4 + 2C_{Q,\beta,d})$. The 2 before $C_{Q,\beta,d}$, different to the exact statement of Theorem 3.2 of Bos & Schmidt-Hieber (2022) is because our $C_{Q,\beta,d}$ is before softmax layer and since softmax layer is Lipchetz of constant 2. We further have $C_1 + 1 \leq 4(5 + 2C_{Q,\beta,d})$, we have

$$
\inf_{\boldsymbol{p} \in \mathcal{F}_\sigma} R(\boldsymbol{p}_0, \boldsymbol{p}) \leq 8C2^{3+\alpha} \frac{(5 + 2C_{Q,\beta,d})^3}{M^{1+\alpha}}\left(1 + \frac{I_{\alpha<1}}{1-\alpha} + \log(M)\right) \lesssim \phi_n \log(n)
$$

Together with oracle bound Equation (35) and $s \asymp n\phi_n \log(n)$, the statement follows. □

**Lemma C.12** (Lemma 3.4 of Bos & Schmidt-Hieber (2022)). *For any $B \geq 2$, $P, Q$ being probability measure on the same measure space, we have*

$$
H^2(P, Q) \leq \frac{1}{2} KL_B(P, Q)
\tag{36}
$$

*where for discrete probabilities*

$$
H^2(P, Q) = \sum_j (\sqrt{P_j} - \sqrt{Q_j})^2
$$

**Remark C.13** (Connecting probability to reward). Since we have $(\sqrt{a} - \sqrt{b})^2 = (a-b)^2/(\sqrt{a} + \sqrt{b})^2$, use Lemma C.12, indicates that in large subset of the embedding space

$$\left| p_0(\boldsymbol{\Psi}_1, \boldsymbol{\Psi}_2) - \hat{p}(\boldsymbol{\Psi}_1, \boldsymbol{\Psi}_2) \right| \lesssim \left| \sqrt{p_0} + \sqrt{\hat{p}} \right| \sqrt{\phi_n L} \log(n)$$

$$\left| r(\boldsymbol{\Psi}_1) - r(\boldsymbol{\Psi}_2) - (\hat{r}(\boldsymbol{\Psi}_1) - \hat{r}(\boldsymbol{\Psi}_2)) \right| \lesssim \frac{\left| \sqrt{p_0} + \sqrt{\hat{p}} \right|}{\tilde{p}(1 - \tilde{p})} \sqrt{\phi_n L} \log(n)$$

where $\tilde{p}$ is a probability between $p_0$ and $\hat{p}$, the second line is due to mean value theorem. This indicates that the comparison should be between those pairs that relatively close in reward to avoid diverging behavior of logit function.

## D  ANALYZING ORDER CONSISTENCY

**Proposition 3.3** (Lower bound on population level order consistency ). *Suppose a learned model $\hat{H}$ achieves objective equation 9 up to $1 - \delta\epsilon$ error for some small $0 < \delta < 1$ and $\epsilon < 3/20$, i.e.,*

$$\mathbb{E}_{x_1, x_2, y_1, y_2, h} \mathbb{1}\left[ h = \hat{H} \right] \geq 1 - \delta\epsilon \tag{10}$$

*Then, with probability at least $1 - \delta$ over $\Delta r$, for any given $\Delta r$ the order consistency of $\hat{r}$ with respect to the oracle utility is bounded below by:*

$$\mathbb{E}_{x_1, x_2, y_1, y_2} \left[ \mathbb{1}\left( \hat{H} \cdot [r(x_1, y_1) - r(x_2, y_2)] \geq 0 \right) \middle| \Delta r \right] \geq (1 - \epsilon) \cdot \xi^2(\Delta r) + \epsilon \cdot (1 - \xi(\Delta r))^2 \tag{11}$$

*Further if we assume that with probability at least $1 - \kappa$, that $\xi(\Delta r) \geq \sqrt{\epsilon^2 + 1 - 3\epsilon} + \epsilon$, we have*

$$\mathbb{E}_{x_1, x_2, y_1, y_2 \sim \ell(x)} \left[ \mathbb{1}\left( \hat{H} \cdot [r(x_1, y_1) - r(x_2, y_2)] > 0 \right) \right] \geq 1 - 4\epsilon - \kappa - \delta \tag{12}$$

*Proof.* The idea of the proof is to first use Markov's inequality to bound probability that for a given distance $\Delta r$ the preference model not well approximate the annotator and under the event that the preference model approximate the annotator well, we bound the total error combined by preference model and annotator.

By assumption we have the (marginal) error probability averaging over all distances $r$ is

$$\mathbb{P}_{x, y_1, y_2, h}\left[ \mathbb{1}\left( \hat{H} \neq h \right) \right]$$
$$= \mathbb{E}_{\Delta r}\left[ \mathbb{P}\left( \hat{H} \neq h \middle| \Delta r \right) \right] \tag{37}$$
$$< \delta\epsilon$$

Denote the random variable

$$\Pi_r := \mathbb{P}\left( \hat{H} \neq h \middle| \Delta r \right) \tag{38}$$

by Markov's inequality

$$\mathbb{P}_r\left( \Pi_r \geq \epsilon \right) \leq \frac{\delta\epsilon}{\epsilon} = \delta \tag{39}$$

In the event $\{\Delta r : \mathbb{P}\left( \hat{H} \neq h \middle| \Delta r \right) < \epsilon\}$, with probability $1 - \delta$ we bound the error rate as function of $\Delta r$. Condition on $\Delta r$, define the following probabilities:

- $p_{\text{annotator}} = \xi(\Delta r)$ is the probability that the annotator $h$ is correct (i.e., agrees with the oracle utility) given the oracle distance.

- $1 - p_{\text{annotator}} = 1 - \xi(\Delta r)$ is the probability that the annotator $\mathcal{H}_\beta$ is incorrect given the oracle distance.

Given the bounded difference between $\hat{H}_\theta$ and $H_\beta$:

- Correct Case: When the annotator is correct, the learned model agrees with the annotator with probability at least $1 - \epsilon$. Thus:

$$p_{\text{correct}} \geq (1 - \epsilon) \cdot \xi(\Delta r). \tag{40}$$

- Incorrect Case: When the annotator is incorrect, the learned model agrees with the annotator with probability at most $\epsilon$. Thus:

$$p_{\text{incorrect}} \leq \epsilon \cdot (1 - \xi(\Delta r)). \tag{41}$$

The order consistency of the learned model $\hat{H}_{\theta^*}$ with the oracle utility can be expressed as:

$$\mathbb{E}_{x,y_1,y_2 \sim \ell(x)}\left[\mathbb{1}\left(\hat{H}(r(y_1, x) - r(y_2, x)) \geq 0\right)\bigg|\Delta r\right] = p_{\text{correct}} \cdot p_{\text{annotator}} + p_{\text{incorrect}} \cdot (1 - p_{\text{annotator}}). \tag{42}$$

Substituting the bounds and simplifying, we have

$$\mathbb{E}_{x,y_1,y_2 \sim \ell(x)}\left[\mathbb{1}\left(\hat{H}(r(y_1, x) - r(y_2, x)) \geq 0\right)\bigg|\Delta r\right] \geq (1 - \epsilon) \cdot \xi^2(\Delta r) + \epsilon \cdot (1 - \xi(\Delta r))^2. \tag{43}$$

Second part of the proof is by observing that $\xi(\Delta r) \geq \sqrt{\epsilon^2 + 1 - 3\epsilon} + \epsilon$ implies $(1 - \epsilon) \cdot \xi^2(\Delta r) + \epsilon \cdot (1 - \xi(\Delta r))^2 \geq 1 - 4\epsilon$ when $\epsilon < 3/20$, consider this conditional bound

$$\mathbb{E}_{x,y_1,y_2 \sim \ell(x)}\left[\mathbb{1}\left(\hat{H}(r(y_1, x) - r(y_2, x)) \geq 0\right)\bigg|\Delta r\right] \geq 1 - 4\epsilon \tag{44}$$

the stated bound fail either 1) $\Pi_r > \epsilon$, with probability at most $\delta$ or 2) $\xi(\Delta r) < \sqrt{\epsilon^2 + 1 - 3\epsilon} + \epsilon$ with probability at most $\kappa$, thus the stated bound is true with probability at least $1 - \kappa - \delta$ due to union bound on failure modes.

Then by definition of conditional probability, the bound in theorem true.

$\square$

**Proposition D.1** (Classification reward). *Suppose data actually coming from BT model Equation (1), and the score $s_i := \text{logit } P(i \text{ wins})$ is connected to BT reward that for a constant $C$ does not depends on $i$*

$$s_i \geq r_i - C$$

*Proof.* We condition on which $j$ that $i$ competed with and apply Jensen's inequality

$$\mathbb{P}(i \text{ wins}) = \mathbb{E}_j[\mathbb{P}(i \succ j|j)] = \mathbb{E}_j\left[\frac{u_i}{u_i + u_j}\right] \geq \frac{u_i}{u_i + \mathbb{E}[u_j]}$$

With some straightforward algebra, we have that

$$\frac{\mathbb{P}(i \text{ wins})}{1 - \mathbb{P}(i \text{ wins})}\mathbb{E}[u_j \geq u_i]$$

Take log at each side and substitute $u_i = \exp(r_i)$ then rearrange

$$s_i := \text{logit } \mathbb{P}(i \text{ wins}) \geq r_i - \log \mathbb{E}[\exp(r_j)]$$

we have $\log \mathbb{E}[\exp(r_j)]$ is a constant. $\square$

## E EXPERIMENT DETAILS

***To enhance the reproducibility of our work, all code, datasets (demonstrations), fine-tuned LLMs, generated training and test responses, annotations of those responses, and their embeddings will be made publicly available.***

**Computational Efficient Experiment Design and Reproducibility**   Our experiments are conducted on a cluster having `128 Intel(R) Xeon(R) Platinum 8336C CPUs @2.30GHz` with `NVIDIA V100 32GB` or `NVIDIA A100 80G` GPU nodes.   We use vllm (Kwon et al., 2023) to accelerate the LLM generation process.

To reproduce our 12,000 experiment results (since we will release the embeddings of the generated responses), only CPUs are needed, and to reproduce all experiments, 6000h CPU-core hours are needed — on a machine with 128 CPU cores, it will take 50 hours to reproduce all of our 12000 experiments (This includes our 5 repeated runs using different random seeds. Running with only 1 seed will take less than 10 hours). Each set-up take less than 30 min CPU-core time usage — less than 1 minutes to finish on a 128-core server.

**Supervised Fine Tuning Stage of Base Models**   Following Stiennon et al. (2020); Bai et al. (2022a); Sun & van der Schaar (2024), we use held-out demonstration datasets generated by GPT4 on the two tasks to conduct SFT on the three base models we use (Gemma2b, Gemma7b, LLaMA3-8b).   Such an SFT procedure generates three SFT-ed checkpoints as additional base LLMs, the Gemma2b-SFT, Gemma7b-SFT, LLaMA3-8b-SFT. The SFT takes less than 10 hours (4 hours for the 2b models) using `A100` GPUs and the TRL framework (von Werra et al., 2020).

**Training and Test Data (Responses) Creation**   The `Harmless` dataset contains 41876 training prompts and 2273 test prompts. The `Helpful` dataset contains 42846 training prompts, and 2292 test prompts.   In our experiment, for each of the 6 base LLMs, we create 10 responses on each training prompt as candidates for reward model training. For each of the test prompts, we create 500 responses and annotate their golden utilities using the golden reward models for testing.

**Creating Embeddings**   We use Gemma2b (Team et al., 2024) to generate embeddings for reward modeling in our experiments. Since we have 6 base LLMs to generate responses and 2 datasets, creating the embeddings on those generations requires GPUs. We use `V100` GPUs with 32GB memory for embedding generation. Each setting (40000 prompts with 10 generations and 2000 prompts with 500 generations) takes around 16 GPU hours to finish.

**Simulated Preference Annotation with Golden Reward Models**   To simulate the imperfect annotation process of human labors, we consider label noises in our experiments following the literature (Ziegler et al., 2019; Dubois et al., 2024; Coste et al., 2023). However, instead of randomly inject noise to the labeling process, we consider a more realistic annotation simulation using the cognitive bottleneck models studied in psychology (Stewart et al., 2005; Guest et al., 2016): the comparisons made between responses that has similar scores will have a higher possibility of being mis-labeled, formally, we have (equation 8)

$$\mathbb{P}\left(h(x, y_1, y_2)(r(x, y_1) - r(x, y_2)) > 0 \middle| \Delta r\right) = \xi(\Delta r),$$

We instantiate $\xi(\Delta r)$ as $\sigma(\beta \Delta r)$, the sigmoid function in determining the probability of getting a correct lable. The $\beta$ parameter here controls the annotation quality: when $\beta = 0$, annotations are purely random, while when $\beta \to \infty$, annotations are perfect. In our experiments, the default setting of $\beta$ is 1 unless explicitly specified otherwise (as in the secton of experiments studing performance under different noise levels in annotations.)

**Hyper-Parameters of the LGB models and MLPs**   To maximally isolate the source of gains, and separate the contribution of methods from their implementations, we use the identical hyper-parameter set up for all experiments (a single default hyper-parameter setup for LightGBM, same MLP configurations for BT and Classification).

For LGB models, we use the default hyper-parameter setting of

```
hyper-param-lgb = {
                'objective': 'binary',
                'metric': 'binary_logloss'
                }
```

For MLPs, we use a minimalism three-layer-feed-forward structure of

```
hyper-param-mlp = {
                'activation': 'ReLU',
                'units': '(1024, 512, 1)',
                'loss': 'BCELoss',
                'optimizer': 'Adam',
                'lr': '0.001',
                'early_stop_patience': '3',
                'max_epoch': '30'
                }
```

While further sweeping on hyper-parameters for each set up will very likely be able to further improve the performance, those engineering efforts are irrelevant of the scope of our research focus on investigating different methods nor informative in drawing conclusions to answer the above questions.

## F  ADDITIONAL RESULTS AND DISCUSSION

**Figures reporting results on changing annotation qualities.** Figure 7.

**Figures reporting results on changing annotation qualities under different annotation availability.** Figure 8 and Figure 9.

**Figures reporting results on changing annotation availability under different annotation qualities.** Figure 10 and Figure 11.

**Figures reporting cross-prompt annotation results on changing annotation availability and annotation quality.** Figure 12 to Figure 15

**Post-hoc explanation of the superiority of the classification-based model over the BT model**

(1). Analytically, the BT model can be used to estimate the *winning rate of each player when matched with another*. On the other hand, the classification-based reward model estimates the *marginal probability of winning*. As we showed in Section 3, both options achieve order consistency, therefore, they both satisfied the minimal requirement for inference time optimization, despite the classification-based score estimation having less information due to the marginalization.

However, given the same dataset, a more informative objective may not be as easy to learn as compared to the less informative objective.

In reward modeling settings, we will have to estimate/learn these logits from *noisy (binary) annotations*. The other side of the same coin (of having less information) in this marginalization/average is that it makes the marginal probability and classification-based reward score easier to learn from data. i.e., we are trading some information that might not be necessary for inference time optimization for a simpler target during the learning/reward modeling stage. We can see this by evaluating the variance of these two targets and one can see that the classification-based reward score has a lower variance, and therefore, an easier-to-learn objective.

$$
\begin{aligned}
Var_{i,j}(P(i \text{ wins } j|i,j)) &= Var_i(\mathbb{E}_j(P(i \text{ wins } j|i,j)|j)) + \mathbb{E}_i(Var_j(P(i \text{ wins } j|i,j)|j)) \\
&= Var_i(P(i \text{ wins }|i)) + \mathbb{E}_i(Var_j(P(i \text{ wins } j|i,j)|j))
\end{aligned}
\tag{45}
$$

The first term on RHS is the classification-based target's variance and the second term is non-negative. So the classification-based method targets an objective with less noise in it.

(2) Empirically, such an insight can be observed in experiments. In Figure 7, we report the results in barplots on changing the annotation quality. - In the top 3 panels, the annotation noises are high (having around or above 30 percent error rate), and in those setups, we find the classification-based reward models in general outperform the BT reward models. - In the bottom 3 panels, the annotation noises are low (less than 15 percent error rate), and in those setups, we find the BT models achieve better or on-par performance as compared with the classification-based reward models.

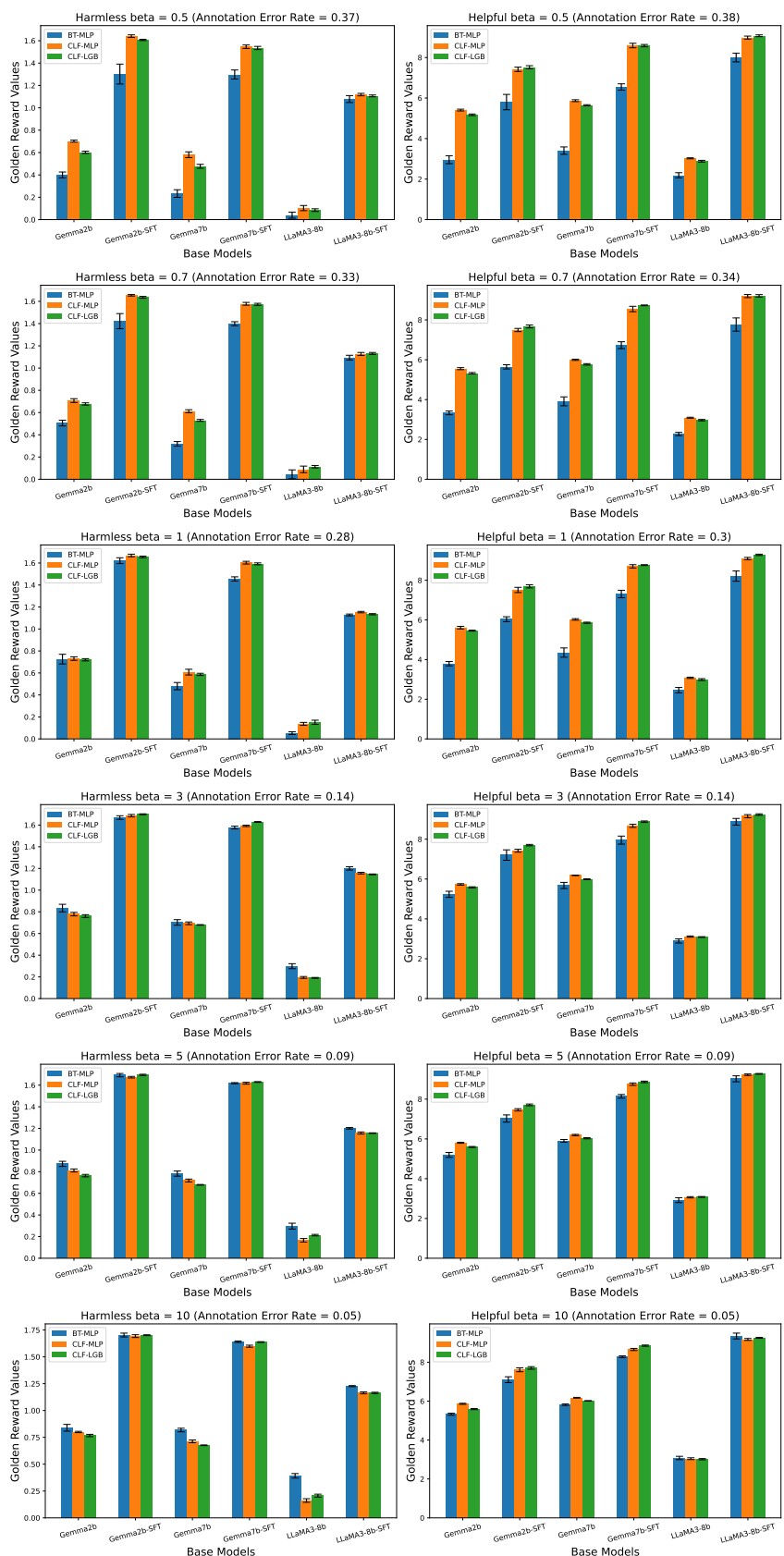

Figure 7: Experiment results in barplots on changing the annotation quality. Error bars are given by 5 runs by changing seeds.

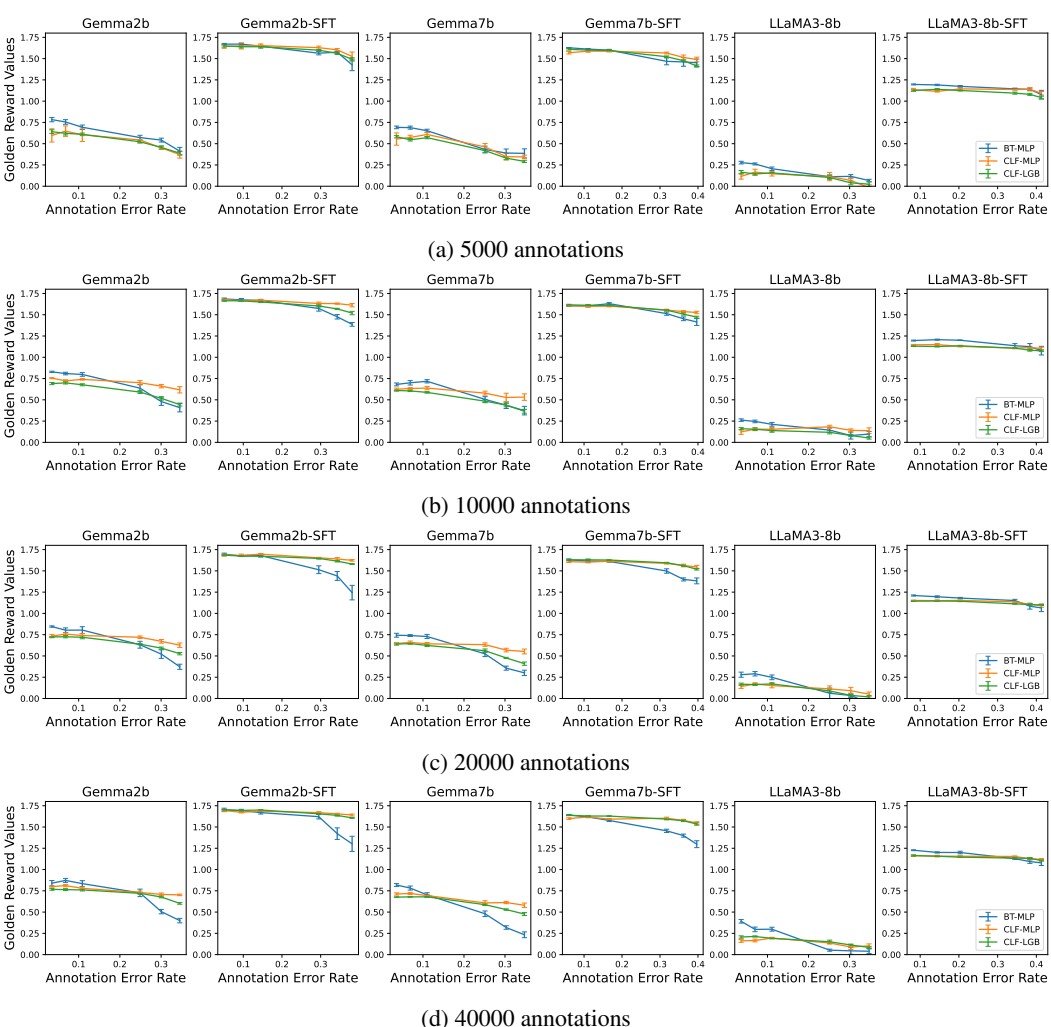

Figure 8: Harmless Dataset: additional results on changing annotation quality under different annotation availability. Error bars are given by 5 runs by changing seeds.

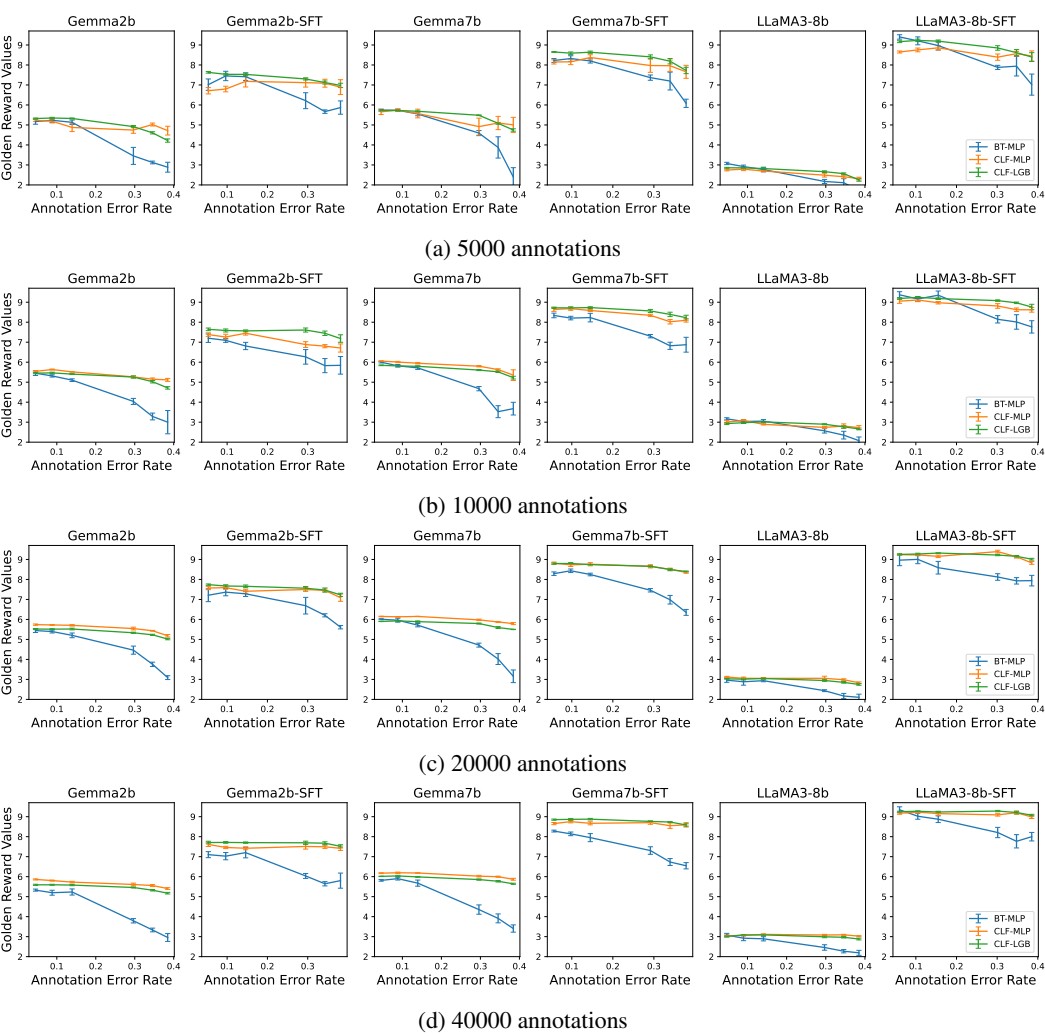

Figure 9: Helpful Dataset: additional results on changing annotation quality under different annotation availability. Error bars are given by 5 runs by changing seeds.

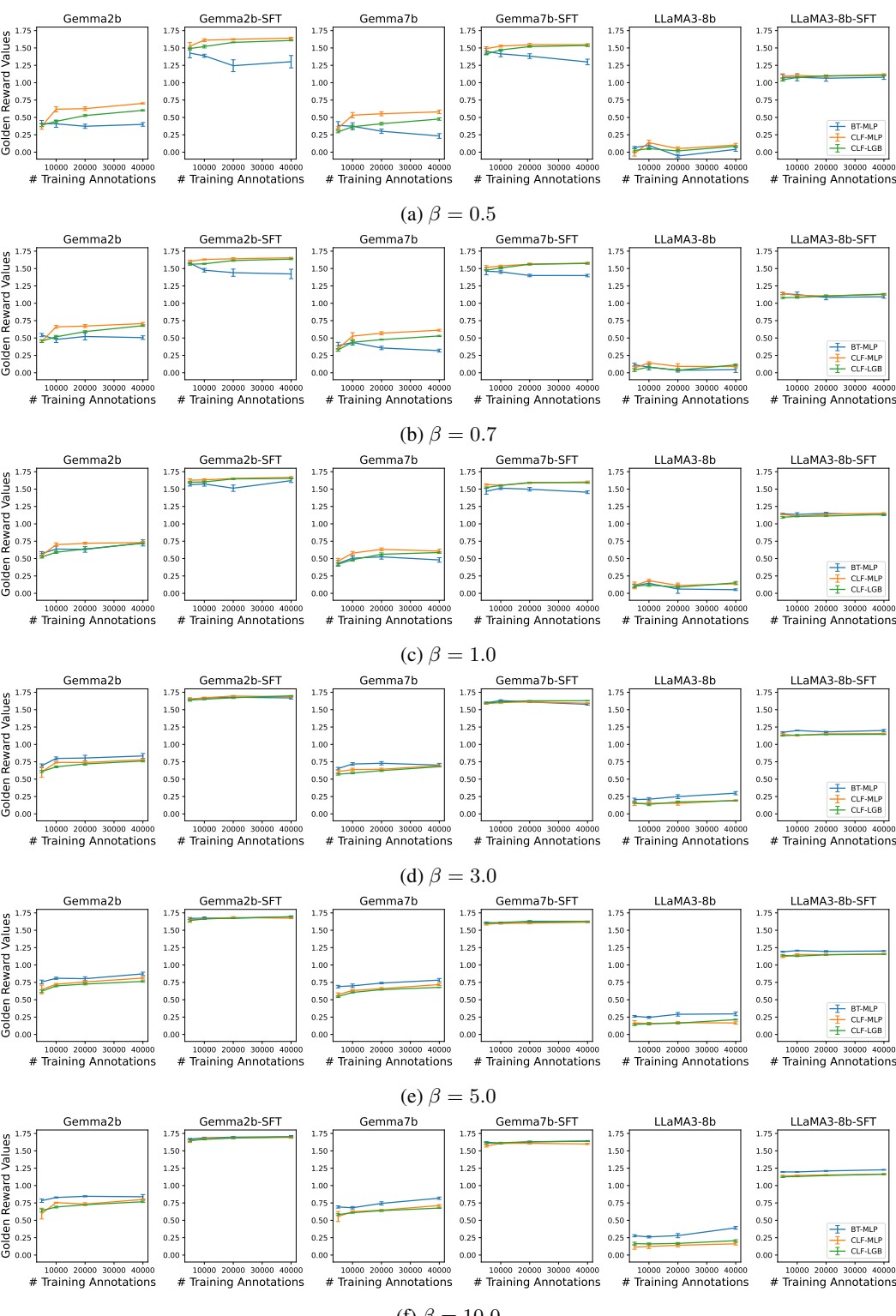

Figure 10: Harmless Dataset: additional results on changing annotation availability under different annotation quality. Error bars are given by 5 runs by changing seeds.

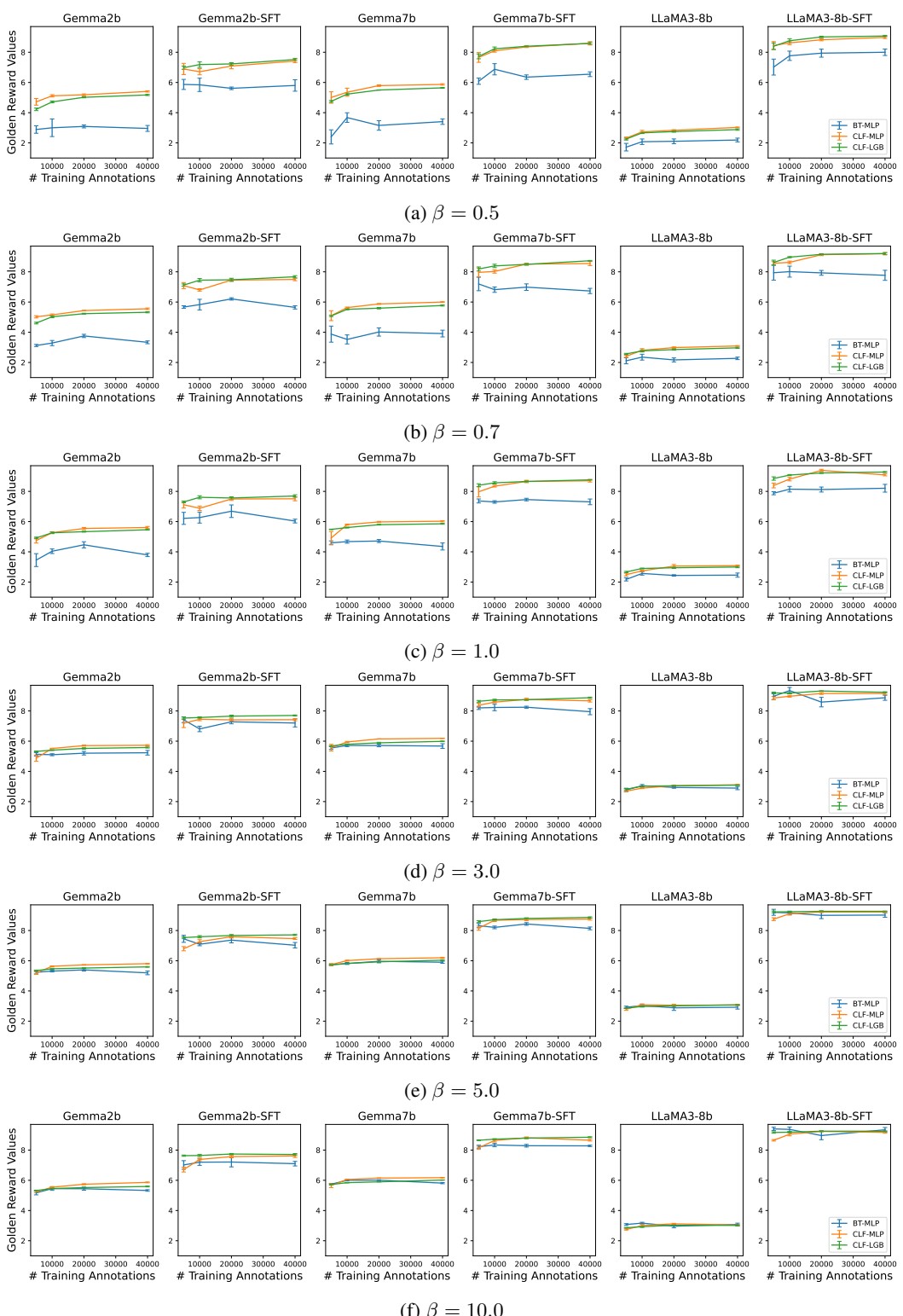

Figure 11: Helpful Dataset: additional results on changing annotation availability under different annotation quality. Error bars are given by 5 runs by changing seeds.

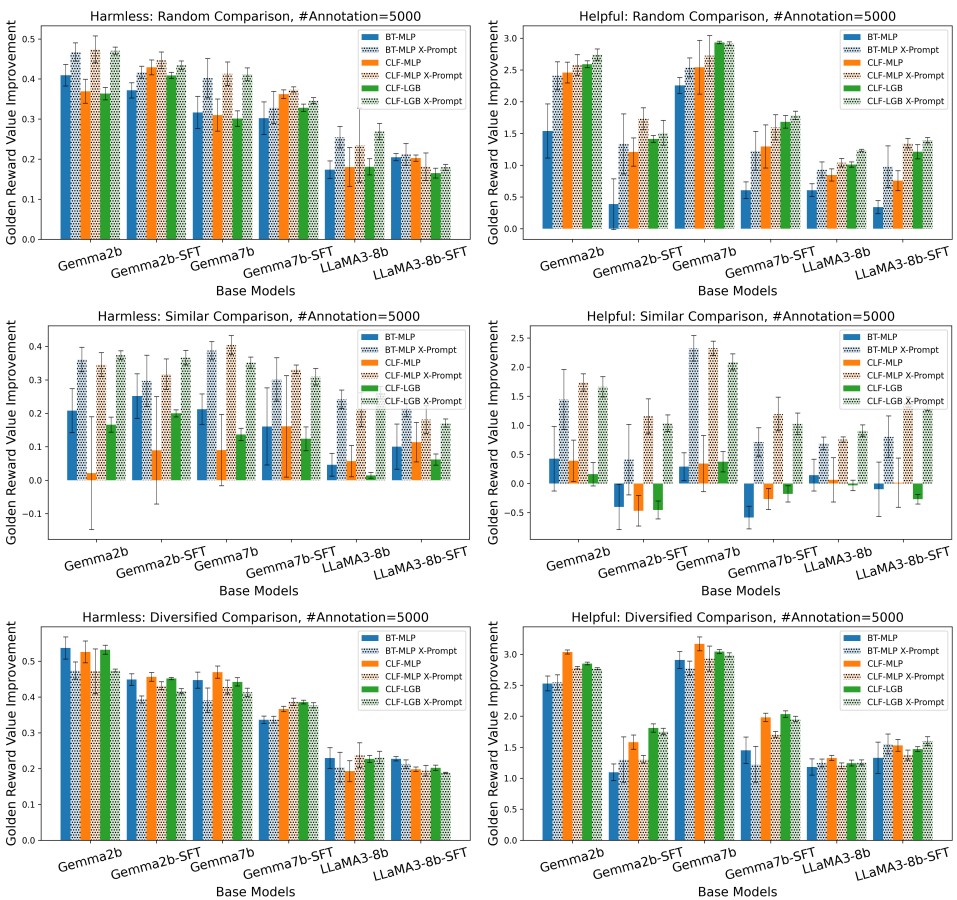

Figure 12: Results on cross-prompt comparisons, with 5000 annotations, $\beta = 1$. Error bars are given by 5 runs by changing seeds.

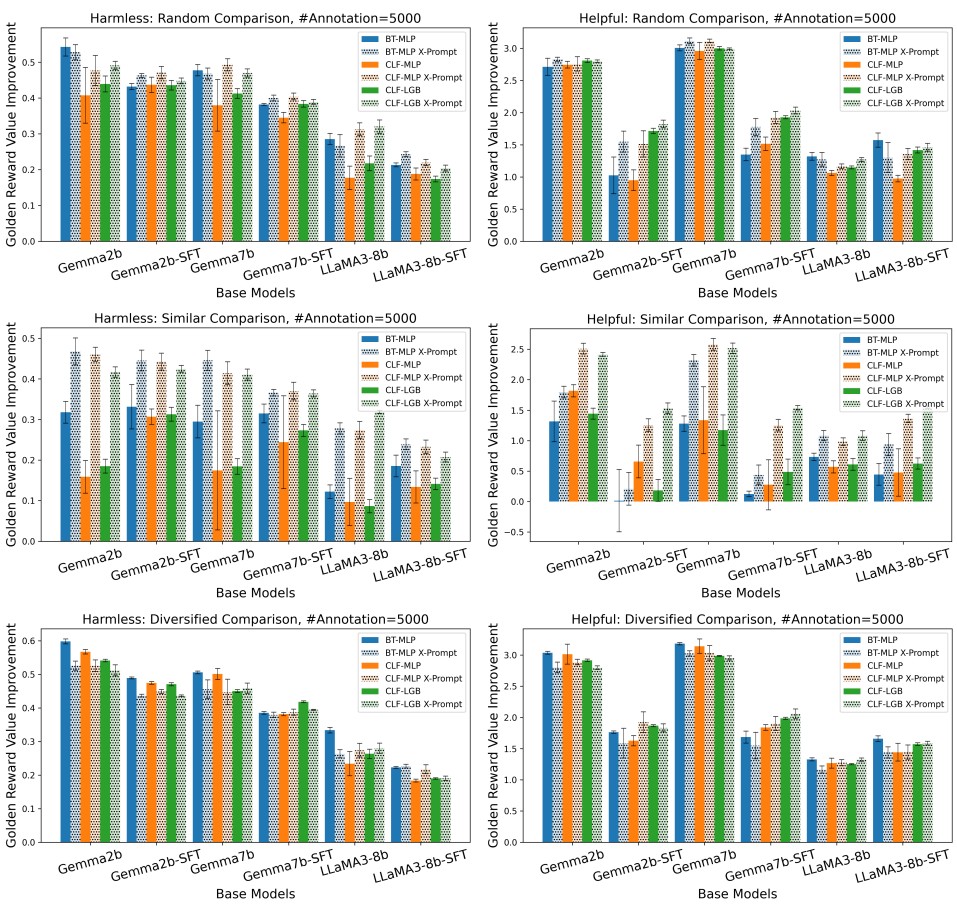

Figure 13: Results on cross-prompt comparisons, with 5000 annotations, $\beta = 10$. Error bars are given by 5 runs by changing seeds.

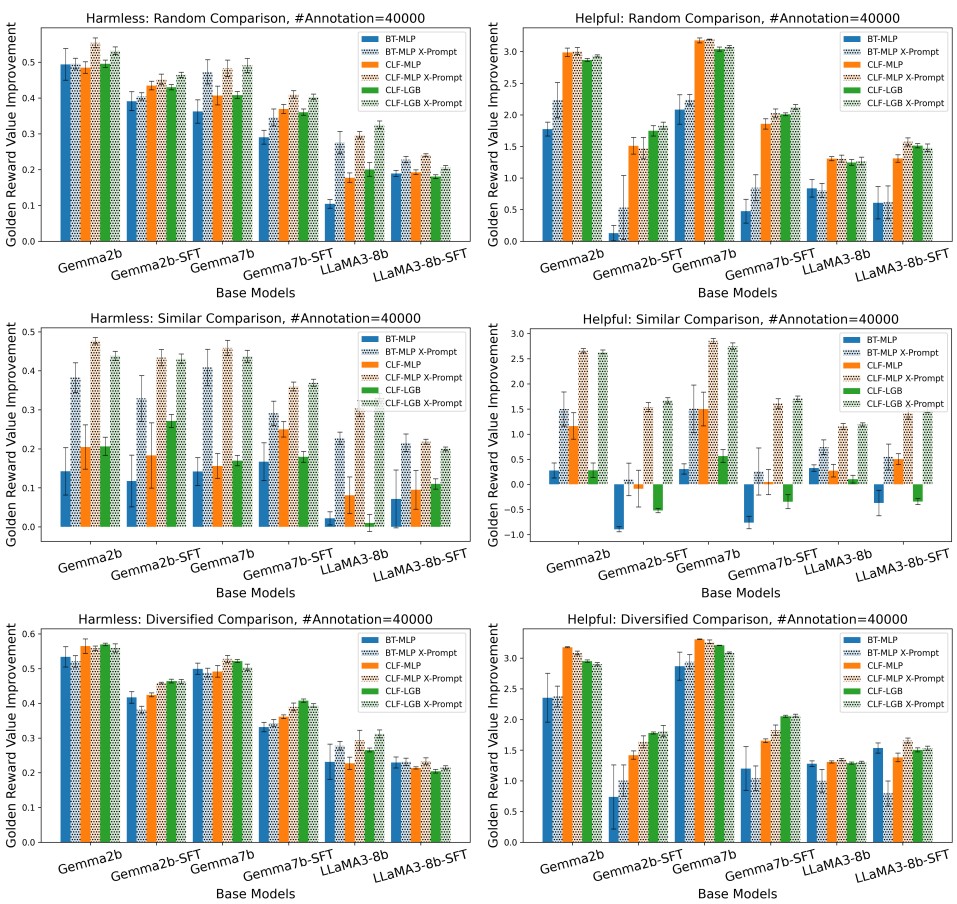

Figure 14: Results on cross-prompt comparisons, with $40000$ annotations, $\beta = 1$. Error bars are given by 5 runs by changing seeds.

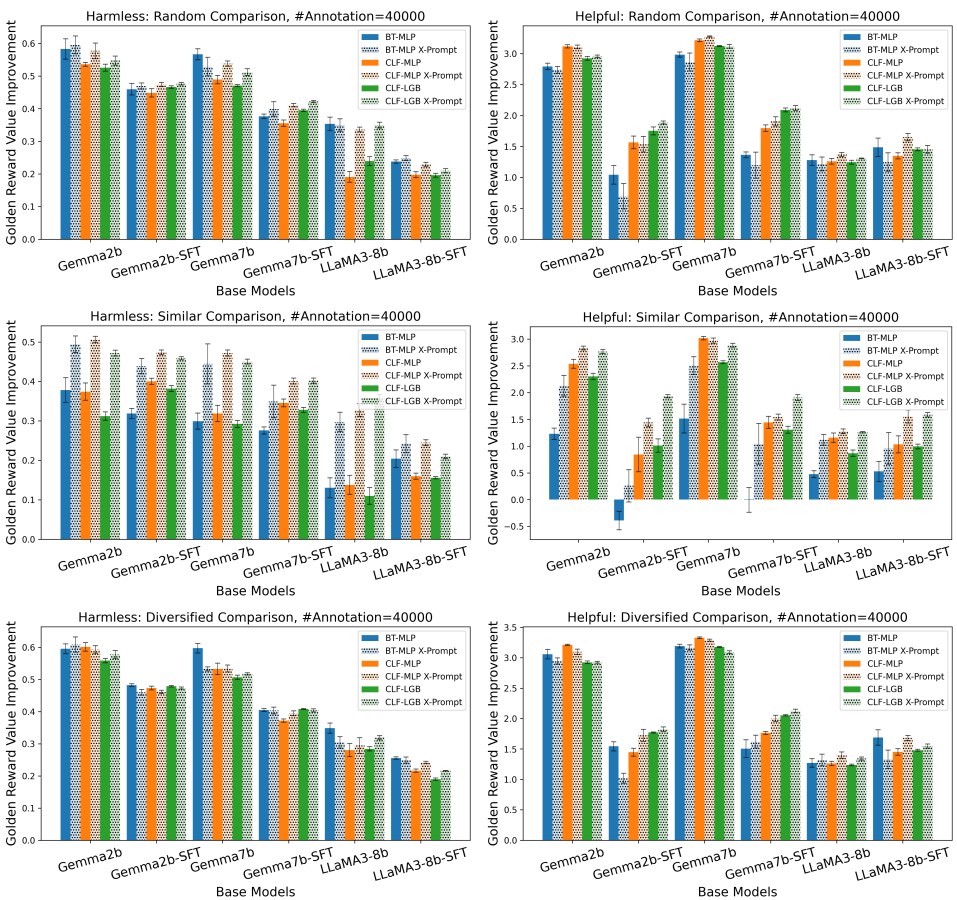

Figure 15: Results on cross-prompt comparisons, with 40000 annotations, $\beta = 10$. Error bars are given by 5 runs by changing seeds.

## G    THEORETICAL JUSTIFICATIONS FOR CROSS-PROMPT COMPARISONS

In both BT and classification-based reward modeling, there is no theoretical requirement to limit comparisons to the same prompts. For classification models, this is straightforward, as they do not rely on paired data at all. Similarly, in traditional BT applications, random pairwise comparisons among players are common. This further motivates our investigation into how randomized comparisons across different prompts affect reward modeling performance.

To further motivate the usage of cross-prompt comparison, we introduce the following notation on annotation quality and analysis as a case study under a Gaussian assumption on score distributions. In equation 8, we consider a special case of $\xi$ in equation 8 to be $\xi(\cdot) = \sigma(\beta \cdot)$, that the annotators' ability is characterized by $\beta$. When $\beta = 0$, we have **random annotations**:

$$\mathbb{P}\left(h(x_1, x_2, y_1, y_2)(r(x_1, y_1) - r(x_2, y_2)) > 0 \Big| \Delta r\right) = \sigma(\beta \Delta r) \xrightarrow{\beta \to 0} 0.5, \qquad (46)$$

and when $\beta \to \infty$, we have **perfect annotations**:

$$\mathbb{P}\left(h(x_1, x_2, y_1, y_2)(r(x_1, y_1) - r(x_2, y_2)) > 0 \Big| \Delta r\right) = \sigma(\beta \Delta r) \xrightarrow{\beta \to \infty} 1. \qquad (47)$$

In a nutshell, the annotator's abilities and the differences among prompt-response pairs together determine how much preference is correctly labeled in the annotation. In the following, we show a special case when two responses of a prompt $x$ are randomly sampled from a single LLM $\ell$.

**Example 1** (Annotation Quality under Gaussian Score). *When data for pairwise annotation is generated through random sampling of two responses $y_1, y_2 \sim \ell(x)$, we further assume the utility of those two responses are sampled from a Gaussian distribution with variance $\sigma_x^2$, i.e., $y \sim \ell(x), r(x, y) \sim \mathcal{N}(\mu_x, \sigma_x^2)$. Then the annotation quality $\mathcal{Q}_{\text{pair}}(x)$ on such a prompt can be defined as the averaged annotation order consistency:*

$$\mathcal{Q}_{\text{pair}}(x) = \mathbb{E}_{y_1, y_2 | x}[\tau_x] = \mathbb{E}_{y_1, y_2 | x}\left[\sigma(\beta | r(x, y_1) - r(x, y_2)|)\right] \qquad (48)$$

*where $\tau_x = \sigma(\beta | r(x, y_1) - r(x, y_2)|)$ is a random variable (over $y_1, y_2$) and the probability density function of $\tau_x$ is*

$$f_{\tau_x | x}(t) = \frac{1}{\sqrt{\pi \beta^2 \sigma_x^2}} \exp\left(-\frac{\left(\log\left(\frac{t}{1-t}\right)\right)^2}{4\beta^2 \sigma_x^2}\right) \cdot \frac{1}{t(1-t)} \qquad (49)$$

*Proof.* To get the PDF of $f(\tau)$, denote $\tau = \sigma(|\rho|)$, where we use $\rho(x) = \beta(r^*(y_1 | x) - r^*(y_2 | x))$, we have $\rho(x) \sim \mathcal{N}(0, 2\beta^2 \sigma_x^2)$ follows the normal distribution, and $|\rho(x)|$ follows a folded normal distribution, its cumulative distribution function, mean and variance are given by

$$F_{|\rho|}(x; \mu = 0, \sigma^2 = 2\beta^2 \sigma_x^2) = \text{erf}\left(\frac{x}{2\beta \sigma_x}\right), \quad \mu_{|\rho|} = \frac{2\beta \sigma_x}{\sqrt{\pi}}, \quad \sigma_{|\rho|}^2 = 2\beta^2 \sigma_x^2(1 - \frac{2}{\pi}), \qquad (50)$$

respectively.

To find the PDF of $\tau = \sigma(|\rho|)$, we use the change of variables formula. If $Y = g(X)$ and $g$ is a monotonic function, the PDF of $Y$, $f_Y(y)$, can be obtained by:

$$f_Y(y) = f_X(g^{-1}(y)) \left| \frac{d}{dy} g^{-1}(y) \right|. \qquad (51)$$

For the sigmoid function, the inverse is given by:

$$g^{-1}(y) = \log\left(\frac{y}{1-y}\right). \qquad (52)$$

The derivative of the inverse sigmoid function is:

$$\frac{d}{dy} g^{-1}(y) = \frac{1}{y(1-y)}. \qquad (53)$$

Plugging these into the change of variables formula, we get:

$$f_\tau(t) = \frac{1}{\sqrt{\pi \beta^2 \sigma_x^2}} \exp\left(-\frac{\left(\log\left(\frac{t}{1-t}\right)\right)^2}{4\beta^2 \sigma_x^2}\right) \cdot \frac{1}{t(1-t)}. \tag{54}$$

$\square$

In this special case, it is easy to get some numerical results: when $\beta^2 \sigma_x^2 = 1$, we have $\mathcal{Q}_{\text{pair}} \approx 0.6749$, e.g., roughly 67.5% of data are correctly labelled by annotators. Similarly, when $\beta^2 \sigma_x^2 = 2$, $\mathcal{Q}_{\text{pair}} \approx 0.7251$; when $\beta^2 \sigma_x^2 = 4$, $\mathcal{Q}_{\text{pair}} \approx 0.7781$; when $\beta^2 \sigma_x^2 = 10$, $\mathcal{Q}_{\text{pair}} \approx 0.8428$; This suggests that the effect of better annotators and the effect of response utility diversity are always coupled: in order to improve data quality, we may either improve the ability of annotators or further **diversify the generation utilities** — as both of those parameters control the annotation quality. Next, we show that cross-prompt comparison in this example can be an effective practice to increase response utility diversity. And show that cross-prompt annotation can improve annotation quality.

**Cross-Prompt Annotation Improves Quality under Gaussian Score**   When considering multiple prompts $x_i, i = 1, 2, ..., N$, we denote the corresponding responses as $y_i$, and scores $r(x_i, y_i) \sim \mathcal{N}(\mu_i, \sigma_i^2)$. In the following, we show that cross-prompt annotation can improve annotation quality.

**Proposition G.1** (Cross-Prompt Comparisons Increase Utility Diversity).   *When data for pairwise annotation is generated through random sampling of two responses $y_1, y_2 \sim \ell(x)$, and the utility of those two responses are sampled from a Gaussian distribution with variance $\sigma_x^2$, i.e., $y \sim \ell(x), r_{x,y} \sim \mathcal{N}(\mu_x, \sigma_x^2)$, when there are multiple prompts $x$, we have*

$$\mathbb{E}_x \mathbb{E}_{y_1, y_2 | x}\left[|r_{x,y_1} - r_{x,y_2}|\right] \leq \mathbb{E}_{x_1, x_2} \mathbb{E}_{y_1 | x_1, y_2 | x_2}\left[|r_{x_1,y_1} - r_{x_2,y_2}|\right] \tag{55}$$

*Proof.* Let $x_{ik} \sim \mathcal{N}(\mu_k, \sigma_k^2)$, and $x_{jl} \sim \mathcal{N}(\mu_l, \sigma_l^2)$, with $k \neq l$, then

$$x_{ik} - x_{jl} \sim \mathcal{N}(\mu_k - \mu_l, \sigma_k^2 + \sigma_l^2) \tag{56}$$

The expectation of $|x_{ik} - x_{jl}|$ is given by

$$\mathbb{E}\left[|x_{ik} - x_{jl}|\right] = \sqrt{\sigma_k^2 + \sigma_l^2} \sqrt{\frac{2}{\pi}} \exp\left(-\frac{(\mu_k - \mu_l)^2}{2(\sigma_k^2 + \sigma_l^2)}\right) + |\mu_k - \mu_l| \text{erf}\left(\frac{|\mu_k - \mu_l|}{\sigma_k^2 + \sigma_l^2}\right) \tag{57}$$

as its special case, let $x_{jk} \sim \mathcal{N}(\mu_k, \sigma_k^2)$, then

$$x_{ik} - x_{jk} \sim \mathcal{N}(0, 2\sigma_k^2) \tag{58}$$

$$\mathbb{E}\left[|x_{ik} - x_{jk}|\right] = 2\sigma_k \sqrt{\frac{1}{\pi}} \tag{59}$$

we consider the special case of $\mu_k = \mu_l$:

$$\frac{\mathbb{E}\left[|x_{ik} - x_{jl}|\right]}{\mathbb{E}\left[|x_{ik} - x_{jk}|\right]} \geq 1, \tag{60}$$

the equality holds only if $\sigma_k^2 = \sigma_l^2$. This is because $\frac{2(1+t)^2}{(1+t)^2}$ reaches its only minimum when $t = 1$ and we can let $t = \frac{\sigma_k}{\sigma_l}$. Since $\sqrt{\frac{2}{\pi}} \exp(-x^2) + |x| \text{erf}(|x|)$ is a monotonically increasing function at $x \geq 0$, equation 60 also holds for $\mu_k \neq \mu_l$. $\square$

This gives us an intuitive result that, in expectation, the reward differences between cross-prompt comparisons are larger than the reward differences between prompt-response pairs sharing the same prompt.

More generally, cross-prompt comparisons improve data quality when the utility distribution of different randomly sampled responses given a single prompt is unimodal and symmetric (e.g., Gaussian).

**Theorem G.2** (Cross-Prompt Annotation Improves Annotation Quality). *When data for pairwise annotation is generated through random sampling of two responses $y_1, y_2 \sim \ell(x)$, and the utility of those two responses are sampled from a location-scale family with probability density function $g_x(x) = f((x - \mu_x)/\sigma_x)$ for $f$ being unimodal and symmetric to 0. For any $\xi : \mathbb{R}_+ \to [1/2, 1]$, first order differentiable, monotone increasing and concave, we have*

$$
\begin{aligned}
\mathbb{E}_x[\mathcal{Q}_{\text{pair}}(x)] &= \mathbb{E}_x \mathbb{E}_{y_1, y_2 | x} \left[ \xi(|r_{x, y_1} - r_{x, y_2}|) \right] \\
&\leq \mathbb{E}_{x_1, x_2} \mathbb{E}_{y_1 | x_1, y_2 | x_2} \left[ \xi(|r_{x_1, y_1} - r_{x_2, y_2}|) \right] := \mathbb{E}_{x_1, x_2}[\mathcal{Q}_{\text{cross-prompt}}(x_1, x_2)].
\end{aligned}
\tag{61}
$$

In the above equation, $\mathcal{Q}_{\text{cross-prompt}}(x_1, x_2)$ is defined slightly different from $\mathcal{Q}_{\text{pair}}(x)$ which only takes one prompt as its input. We can understand $\mathcal{Q}_{\text{pair}}(x)$ as a special case of $\mathcal{Q}_{\text{cross-prompt}}(x_1, x_2)$ when $x_1 = x_2 = x$. Theorem G.2 highlights that cross-prompt comparisons improve annotation quality for a broad class of utility distributions and $\xi$.

*Proof.* Theorem G.2 follows directly the combination of Lemma G.3 - Lemma G.5. □

**Section Summarization** In this section, we highlighted the theoretical superiority of using cross-prompt comparisons in preference annotations. In expectation, cross-prompt comparisons can improve annotation quality since they increase the expected differences between prompt-response pairs.

**Lemma G.3.** *Suppose $\xi : \mathbb{R}_+ \to [1/2, 1]$, first order differentiable, monotone increasing and $z \sim f$ with $f$ being density symmetric to 0 and unimodal. We have for all $\mu$*

$$
\mathbb{E}(\xi(|z + \mu|)) \geq \mathbb{E}(\xi(|z|))
\tag{62}
$$

*Proof.* Without loss of generality, we assume $\mu \geq 0$. Suppose the results hold for $\mu \geq 0$, it has to hold for $\mu \leq 0$. To see that, we observe that $-z \sim P$ due to symmetry, and apply the result for positive $\mu$ so that $\mathbb{E}(\xi(|z + \mu|)) = \mathbb{E}(\xi(|-z - \mu|)) \geq \mathbb{E}(\xi(|-z|)) = \mathbb{E}(\xi(|z|))$.

It thus suffices to show the result for nonnegative $\mu$. To do so, we prove that this expectation, as a function of $\mu$, is monotone increasing by taking the derivative.

$$
\begin{aligned}
\frac{d}{d\mu} \mathbb{E}(\xi(|z + \mu|)) &= \mathbb{E} \left[ \frac{\partial}{\partial \mu} \xi(|z + \mu|) \right] \\
&= \mathbb{E} \left[ \frac{d}{d|z + \mu|} \xi(|z + \mu|) \operatorname{sign}(z + \mu) \right] \\
&= \int_{-\infty}^{\infty} \frac{d}{d|z + \mu|} \xi(|z + \mu|) \operatorname{sign}(z + \mu) f(z) dz \\
&= \int_{-\mu}^{\infty} \frac{d}{d|z + \mu|} \xi(|z + \mu|) f(z) dz - \int_{-\infty}^{-\mu} \frac{d}{d|z + \mu|} \xi(|z + \mu|) f(z) dz \\
&= \int_{0}^{\infty} \frac{d}{d|z|} \xi(|z|) f(z - \mu) dz - \int_{-\infty}^{0} \frac{d}{d|z|} \xi(|z|) f(z - \mu) dz \\
&= \int_{0}^{\infty} \frac{d}{d|z|} \xi(|z|) f(z - \mu) dz - \int_{0}^{\infty} \frac{d}{d|z|} \xi(|z|) f(z + \mu) dz \\
&\geq 0
\end{aligned}
$$

The last line due to unimodality and we must have for all $z \in [0, \infty)$ $f(z - \mu) \geq f(z + \mu)$ for $\mu \geq 0$ while $\frac{d}{d|z|} \xi(|z|)$ symmetric to 0 and bounded. □

**Lemma G.4.** *Suppose $x_1, x_2$ iid from a unimodal symmetric location scale family density $g_x$, i.e., the density of $x_1, x_2$ can be written as $g_x(x) = f((x - \mu_x)/\sigma_x)$ for $f$ being unimodal and symmetric to 0. Further, suppose $y_1, y_2 \sim g_y$ iid with density $g_y(y) = f((y - \mu_y)/\sigma_y)$ for the same $f$, we have for a function $\xi : \mathbb{R}_+ \to [1/2, 1]$, first order differentiable, monotone increasing and concave, that*

$$
\frac{1}{2} \mathbb{E}(\xi(|x_1 - x_2|)) + \frac{1}{2} \mathbb{E}(\xi(|y_1 - y_2|)) \leq \mathbb{E}(\xi(|x_1 - y_1|))
\tag{63}
$$

*Proof.* By assumption $y_1, y_2, x_1, x_2$'s are from the same location-scale family. There exists a $z$ whose density is $f$ such that $y_1$ has the same distribution as $\sigma_y z + \mu_y$ and $y_1 - y_2$ having the same distribution as $\sqrt{2}\sigma_y z$, $x_1 - x_2$ having same distribution as $\sqrt{2}\sigma_x z$ and $x_1 - y_1$ having the same distribution with scale $\sqrt{\sigma_x^2 + \sigma_y^2}$ and location $\mu_x - \mu_y$ We first find an upper bound of the left-hand side using Jensen's inequality as $\xi$ concave

$$\frac{1}{2}\mathbb{E}(\xi(|x_1 - x_2|)) + \frac{1}{2}\mathbb{E}(\xi(|y_1 - y_2|)) = \frac{1}{2}\mathbb{E}(\xi(\sqrt{2}\sigma_x|z|)) + \frac{1}{2}\mathbb{E}(\xi(\sqrt{2}\sigma_y|z|)) \tag{64}$$

$$= \mathbb{E}(\frac{1}{2}\xi(\sqrt{2}\sigma_x|z|) + \frac{1}{2}\xi(\sqrt{2}\sigma_y|z|)) \tag{65}$$

$$\leq \mathbb{E}(\xi(\frac{1}{2}\sqrt{2}\sigma_x|z| + \frac{1}{2}\sqrt{2}\sigma_y|z|) \tag{66}$$

$$= \mathbb{E}\left(\xi\left(\sqrt{\frac{(\sigma_x + \sigma_y)^2}{2}}|z|\right)\right) \tag{67}$$

$$\tag{68}$$

The righthand side

$$\mathbb{E}(\xi(|x_1 - y_1|)) = \mathbb{E}\left[\xi\left(|\mu_x - \mu_y + \sqrt{\sigma_x^2 + \sigma_y^2}z|\right)\right] \tag{69}$$

$$\geq \mathbb{E}\left[\xi\left(\sqrt{\sigma_x^2 + \sigma_y^2}|z|\right)\right] \tag{70}$$

by Lemma G.3. We observe that for all $\sigma_x, \sigma_y$ that

$$\sqrt{\frac{(\sigma_x + \sigma_y)^2}{2}} \leq \sqrt{\sigma_x^2 + \sigma_y^2} \tag{71}$$

Thus for all $|z|$, we have $\xi\left(\sqrt{\sigma_x^2 + \sigma_y^2}|z|\right) \geq \xi\left(\sqrt{\frac{(\sigma_x + \sigma_y)^2}{2}}|z|\right)$ and in turn by taking expectation the result in the statement is true. $\square$

**Lemma G.5.** *We have that for random variables described above and suppose $\sigma_x, \sigma_y$ are iid, $\mu_x, \mu_y$ are iid*

$$\mathbb{E}_{\sigma_x, \mu_x}\mathbb{E}_{x|\sigma_x, \mu_x}(\xi(|x_1 - x_2|)) \leq \mathbb{E}_{\sigma_x, \sigma_y, \mu_x, \mu_y}\mathbb{E}_{x, y|\sigma_x, \sigma_y, \mu_x, \mu_y}(\xi(|x_1 - y_1|)) \tag{72}$$

*Proof.* Since $\sigma_x, \sigma_y, \mu_x, \mu_y$ iid we can rewrite the left hand side as

$$\mathbb{E}_{\sigma_x, \mu_x}\mathbb{E}_{x|\sigma_x, \mu_x}(\xi(|x_1 - x_2|)) = \frac{1}{2}\left[\mathbb{E}_{\sigma_x, \mu_x}\mathbb{E}_{x|\sigma_x, \mu_x}(\xi(|x_1 - x_2|)) + \mathbb{E}_{\sigma_y, \mu_y}\mathbb{E}_{y|\sigma_y, \mu_y}(\xi(|y_1 - y_2|))\right] \tag{73}$$

$$= \mathbb{E}_{\sigma_x, \sigma_y, \mu_x, \mu_y}\left[\frac{1}{2}\mathbb{E}_{x|\sigma_x, \mu_x}(\xi(|x_1 - x_2|)) + \frac{1}{2}\mathbb{E}_{y|\sigma_y, \mu_y}(\xi(|y_1 - y_2|))\right] \tag{74}$$

and the statement reduces to the two-pair case we showed before. $\square$

# H    DIRECT EVALUATION METRICS FOR REWARD MODELS

Explicitly evaluating reward models' accuracies would be useful in understanding the performance of different reward models. Below, we present the results of two evaluation metrics: we first show the success rate of choosing a higher-quality response as compared with the population median score of responses (higher is better) in Figure 16. In Figure 17, we present the performance percentile of the reward model selected responses in the population (higher is better). We find different implementations of the classification-based reward models perform similarly and are in general better than using the BT models. And those discoveries are well aligned with the BoN evaluations.

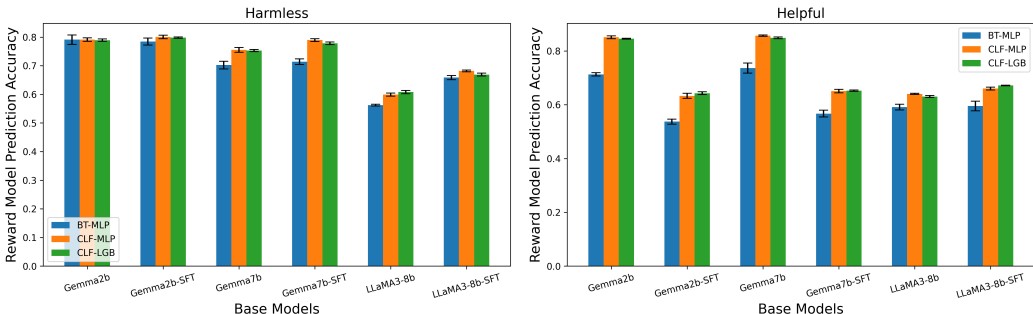

Figure 16: Success rate of choosing a higher-scored response than the population median on the Harmless (left) and Helpful (right) datasets. The results are from 5 seeds.

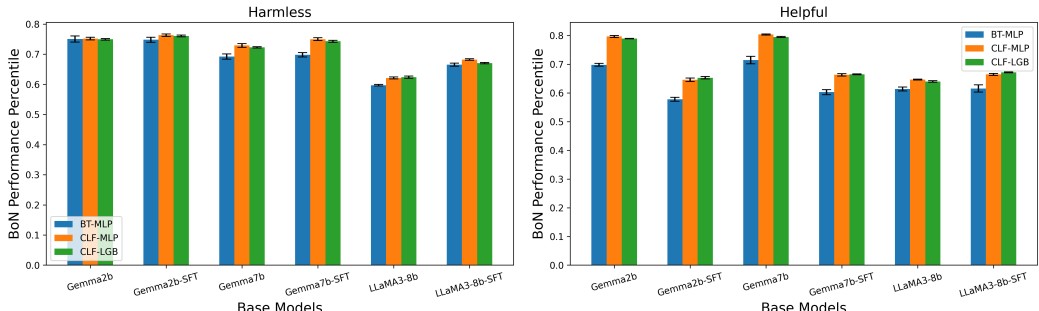

Figure 17: BoN performance percentile on the Harmless (left) and Helpful (right) datasets. The results are from 5 seeds.

