# OpenReview forum: "Rethinking Reward Modeling in Preference-based Large Language Model Alignment"
_ICLR.cc/2025/Conference — ICLR 2025 Oral_

### Official Review · Reviewer_omr2 · 2024-11-03

**Soundness:** 3
**Presentation:** 3
**Contribution:** 3
**Rating:** 8
**Confidence:** 3

**Summary:**

The paper addresses the prevalent use of the Bradley-Terry (BT) model for reward modeling in LLM alignment. It introduces an asymptotic theory for BT regression in LLM reward modeling, setting the first risk bound for BT model in LLM alignment, and propose order consistency as the central goal of reward modeling, which can inform both the BT model and an alternative classification-based approach. This approach is shown to be more flexible and efficient than the BT model, especially when dealing with varying levels of annotation quality and quantity. The paper supports its claims with extensive testing across different LLMs and annotation strategies.

The proposed theoretical framework includes:
- Argues that a reward model does not need to provide exact probabilities of preference; instead, it must maintain the correct order of preferences between prompt-response pairs.
- Establishes a bound on the prediction error of a reward model based on MLPs. This bound ensures that, under specific conditions, the model will predict preference probabilities with a controlled error margin.
- Shows that the predicted rewards will be close to the true rewards as long as the preference probabilities are well approximated.
- Provides a theoretical guarantee that the model will remain order-consistent even when dealing with noisy annotations.

**Strengths:**

- The paper provides a theoretical framework to justify the use of order consistency.
- The authors conduct extensive experiments with diverse setups, support their claims about the classification-based model’s advantages over BT.
- The paper is well written.

**Weaknesses:**

- The paper evaluates reward model performance primarily through BoN sampling improvements. While BoN results provide insights into relative ranking, they do not directly reflect the reward model’s prediction accuracy or error, which is central to the theoretical claims in Theorem 2.2. Including accuracy or error metrics in the evaluation could provide a more comprehensive validation of the theoretical bounds on prediction error, demonstrating that the model’s predictions align with the theoretical guarantees.
- In RL training (e.g., using PPO), the RM’s calibration is crucial: a well-calibrated RM should assign progressively higher rewards to better trajectories. Theorem 2.2 focuses on minimizing prediction error, which may not ensure adequate calibration for RL. To fully evaluate the proposed approach, an analysis of the reward model’s calibration or additional experiments using PPO (or other algorithms) would help determine if the model can effectively support RL tasks where calibration quality directly impacts learning stability and performance.

**Questions:**

- In RLHF, preferences may be non-transitive—such that if A > B and B > C, it doesn’t always imply A > C. Could this be consider as annotation errors? Or should this also be invested (as it might also reflect genuine complexities in human judgments, influenced by context or subjective factors)?

---

> ### Author Response · Authors · 2024-11-18
> **Author Response to Reviewer omr2**
>
> We thank the reviewer for their time and effort devoted to reviewing our paper, and for their encouraging comments and insightful questions, below we would like to respond to each of the points in turn.
>
> ---
>
> ## 1. Direct Evaluation of Reward Model Performance
>
> We thank the reviewer for pointing out that explicitly evaluating reward models’ accuracies would be useful in understanding the performance of different reward models. **We have added the corresponding results in the table below and Appendix H in our revised manuscript.**
>
> In Table 1 and Table 2, we show the success rate of choosing a higher-quality response as compared with the population median score of responses (higher is better). In Table 3 and Table 4, we present the performance percentile of the reward model selected responses in the population (higher is better).
> We find different implementations of the classification-based reward models perform similarly and are in general better than using the BT models. And those discoveries are well aligned with the BoN evaluations.
>
> Table 1. P(Selected > Median) on the Harmless Dataset
>
> | Model           | BT | CLF-MLP | CLF-LGB |
> |-----------------|----------------------|-----------------------|-----------------------|
> | Gemma2b         | 0.791 ± 0.017       | 0.792 ± 0.006        | 0.789 ± 0.004        |
> | Gemma2b-SFT     | 0.784 ± 0.012       | 0.801 ± 0.006        | 0.798 ± 0.003        |
> | Gemma7b         | 0.702 ± 0.013       | 0.755 ± 0.009        | 0.753 ± 0.004        |
> | Gemma7b-SFT     | 0.714 ± 0.01        | 0.789 ± 0.005        | 0.778 ± 0.005        |
> | LLaMA3-8b       | 0.562 ± 0.003       | 0.599 ± 0.005        | 0.608 ± 0.006        |
> | LLaMA3-8b-SFT   | 0.659 ± 0.007       | 0.682 ± 0.003        | 0.669 ± 0.005        |
>
> Table 2. P(Selected > Median) on the Helpful Dataset
>
> | Model           | BT | CLF-MLP | CLF-LGB |
> |-----------------|----------------------|-----------------------|-----------------------|
> | Gemma2b         | 0.713 ± 0.006       | 0.851 ± 0.005        | 0.845 ± 0.001        |
> | Gemma2b-SFT     | 0.537 ± 0.009       | 0.633 ± 0.009        | 0.643 ± 0.005        |
> | Gemma7b         | 0.736 ± 0.019       | 0.857 ± 0.003        | 0.849 ± 0.003        |
> | Gemma7b-SFT     | 0.567 ± 0.012       | 0.651 ± 0.006        | 0.652 ± 0.002        |
> | LLaMA3-8b       | 0.591 ± 0.011       | 0.64 ± 0.002         | 0.63 ± 0.004         |
> | LLaMA3-8b-SFT   | 0.595 ± 0.018       | 0.66 ± 0.005         | 0.671 ± 0.001        |
>
>
> Table 3. BoN Performance Percentile on the Harmless Dataset
> | Model           | BT | CLF-MLP | CLF-LGB |
> |-----------------|----------------------|-----------------------|-----------------------|
> | Gemma2b         | 0.75 ± 0.01         | 0.752 ± 0.004        | 0.749 ± 0.003        |
> | Gemma2b-SFT     | 0.748 ± 0.008       | 0.763 ± 0.004        | 0.761 ± 0.003        |
> | Gemma7b         | 0.692 ± 0.009       | 0.729 ± 0.006        | 0.722 ± 0.002        |
> | Gemma7b-SFT     | 0.698 ± 0.007       | 0.75 ± 0.005         | 0.743 ± 0.003        |
> | LLaMA3-8b       | 0.597 ± 0.003       | 0.621 ± 0.003        | 0.623 ± 0.004        |
> | LLaMA3-8b-SFT   | 0.665 ± 0.005       | 0.682 ± 0.003        | 0.67 ± 0.002         |
>
>
> Table 4. BoN Performance Percentile on the Helpful Dataset
>
> | Model          | BT | CLF-MLP | CLF-LGB |
> |-----------------|----------------------|-----------------------|-----------------------|
> | Gemma2b         | 0.698 ± 0.005       | 0.797 ± 0.003        | 0.79 ± 0.001         |
> | Gemma2b-SFT     | 0.578 ± 0.007       | 0.646 ± 0.006        | 0.653 ± 0.004        |
> | Gemma7b         | 0.715 ± 0.013       | 0.804 ± 0.002        | 0.795 ± 0.002        |
> | Gemma7b-SFT     | 0.603 ± 0.009       | 0.663 ± 0.004        | 0.665 ± 0.002        |
> | LLaMA3-8b       | 0.614 ± 0.007       | 0.647 ± 0.001        | 0.64 ± 0.003         |
> | LLaMA3-8b-SFT   | 0.616 ± 0.013       | 0.665 ± 0.004        | 0.672 ± 0.002        |

---

> > ### Author Response · Authors · 2024-11-18
> > **Author Response to Reviewer omr2 (Cont.)**
> >
> > ## 2. Reward model calibration.
> >
> > _Clarification: with **calibration**, we suppose the reviewer is referring to the property that a higher reward model score should correspond to a higher oracle utility score. Please correct us if we understand the reviewer’s comment incorrectly!_
> >
> > ---
> > We thank the reviewer for pointing out the importance of having calibrated reward models, and we fully agree such a property is essential for the success of the reward modeling process. In our Theorem 2.2, the objective is to demonstrate why the BT model works in estimating reward function, hence the analyses in Section 2 mainly focus on minimizing errors in score predictions.
> >
> > In section 3 of our paper, we discussed the essential property of order consistency (i.e., reward model calibration) in reward modeling — in our work, the order consistency formally defined whether a reward model is well-calibrated, and **we highlighted the objective of reward modeling should be optimization toward better order consistency rather than toward more precise win rate predictions as in the BT models.** This is because the final objective of reward modeling is to optimize LLMs to generate better responses using those reward models, instead of knowing the exact win rate between any two responses. In our work, **we have Proposition 3.3 to theoretically justify the importance of having order-consistent (i.e., calibrated) reward models**, and we explained in section 3.2 and section 3.3 why both the BT models and the classification-based reward models are optimizing toward such an objective.
> >
> > Empirically, the percentile performances we reported above (in Table 3 and Table 4 of the previous response) work as an additional evaluation metric that directly quantifies whether different reward models are properly calibrated (or with the concept defined and used in our paper, order consistent).
> >
> > ---
> > ## 3. Discussion on preference transitivity
> >
> > We thank the reviewer for raising the question and inspiring a discussion of the potential non-transitive preference annotations.
> >
> > In the canonical RLHF framework, the reward modeling step builds a deterministic reward model that assigns a scalar value to each of the responses, and the fundamental assumption is that such an oracle utility value exists. RLHF chose to build reward models from pairwise preference annotations rather than direct score-based annotations due to the consistency challenge in the latter approach.
> >
> > Therefore, the existence of such an oracle utility value lays the foundation of the policy optimization step that seeks to maximize the utility value. An interpretation of the oracle utility value can be the population-level preferences. From such a perspective, **the non-transitive annotations manifest the imperfectness of annotators in identifying the population-level preferences**, or this can be interpreted as an individual-dependent bias from another point of view.
> >
> > Due to space limit, **we deferred an extended discussion linking those individual-level bias and reward modeling in Appendix B (pages 16-17 in the updated manuscript, we have highlighted those contents with orange text).** We also elaborated on the underlying assumptions and motivation for applying the BT models to the preference data annotated by different individuals.
> >
> > To sum up, the **individual annotations** in a preference dataset can contain non-transitive preferences, yet **the reward modeling objective is to learn the population-level average of those imperfect preference annotations**, which is transitive by definition.
> >
> >
> > ---------
> >
> > Once again, we would like to thank the reviewer for their insightful comments and thorough reading to improve our paper. We hope the responses above have addressed the outstanding questions and the reviewer would consider raising their score if those questions have been appropriately answered. Should there be any leftover concerns, please don’t hesitate to let us know and we will do our utmost to address them!

---

> > > ### Comment · Reviewer_omr2 · 2024-11-25
> > >
> > > I appreciate your response. Most of my concerns and questions have been adequately addressed. Consequently, I have decided to raise my rating to an 8.

---

### Official Review · Reviewer_wGfE · 2024-11-03

**Soundness:** 4
**Presentation:** 4
**Contribution:** 3
**Rating:** 8
**Confidence:** 4

**Summary:**

This paper investigates the theoretical and practical foundations of the BT model in LLM alignment, examining its common use in transforming pairwise comparisons into reward scores despite sparse data. The authors establish a convergence rate for BT reward models based on neural network embeddings, ensuring its theoretical validity, but argue that BT is not essential for downstream optimization, as any monotonic transformation of true reward suffices. They propose an order-consistent alternative using a binary classifier framework that simplifies reward modeling. Furthermore, the authors propose to utilize cross-prompt comparisons for improving the reward modeling performance.

**Strengths:**

1. Provides a thorough analysis of the BT model's application in LLM alignment, identifying unique challenges and offering theoretical justification, including the introduction of asymptotic theory for BT regression in preference-based reward modeling.
2. Proposes `order consistency` as a core principle in reward modeling, leading to an alternative to the BT model. Additionally, the use of cross-prompt comparisons enhances training by expanding available comparisons, which further contributing to performance improvements.
3. The paper is presented in a clear and coherent manner. The authors introduce the purpose behind each step before detailing the results, providing a helpful summary of expected outcomes. Despite the dense content, the writing remains accessible and easy to follow.
4. The experiment is thoughtfully designed to validate the proposed methods and offer valuable insights to readers.

**Weaknesses:**

I have no major concerns. I have only one minor suggestion: Figures 2 and 3 should include dataset names in each row for added clarity.

**Questions:**

- **Q1:** The results showing that $\\mathcal{L}_{\\textrm{clf}}$ outperforms the BT model in Figure 1 are somewhat counterintuitive to me, as the classification objective learns marginal probability while the BT model learns joint probability (line 305). The classification objective seems to use less information than the BT model, as the correspondence between $i$ and $j$ is missing. Could you elaborate on why the classification objective not only works but also performs better than the BT model?

- **Q2:** The idea of utilizing cross-prompt comparisons is interesting and shows significant improvements in experiments. Could you discuss any restrictions on using cross-prompt comparisons and describe scenarios where cross-prompt comparisons are particularly effective?

---

> ### Author Response · Authors · 2024-11-18
> **Author Response to Reviewer wGfE**
>
> We thank the reviewer for devoting time to reviewing our paper, and for their encouraging comments and insightful questions, below we respond to each of the points in turn.
>
> ---
> ## 1. Figure captions update
>
> We thank the reviewer for pointing out the missing dataset names, we have added the dataset names on the captions accordingly.
>
> ---
> ## 2. Post-hoc explanation of the superiority of the classification-based model over the BT model
>
> We agree with the reviewer that the marginal probability of one winning a game is a less informative object than the conditional probability of i wins j.
>
>
> (1). **Analytically**, the BT model can be used to estimate the **winning rate of each player when matched with another**. On the other hand, the classification-based reward model estimates the **marginal probability of winning**. As we showed in Section 3, both options achieve **order consistency**, therefore, they both satisfied the minimal requirement for inference time optimization, despite the classification-based score estimation having less information due to the marginalization.
>
> However, given the same dataset, a more informative objective may not be as easy to learn as compared to the less informative objective.
>
> In reward modeling settings, we will have to estimate/learn these logits from **noisy (binary) annotations**. The other side of the same coin (of having less information) in this marginalization/average is that it makes the marginal probability and classification-based reward score easier to learn from data. i.e., we are trading some information that might not be necessary for inference time optimization for a simpler target during the learning/reward modeling stage. We can see this by evaluating the variance of these two targets and one can see that the classification-based reward score has a lower variance, and thus as the reviewer correctly pointed out a less informative, but easier-to-learn object.
>
> $Var_{i,j}(P(i \textrm{ wins } j|i,j)) = Var_i(\mathbb{E}_j(P(i \textrm{ wins } j|i,j)|j)) + \mathbb{E}_i(Var_j(P(i \textrm{ wins } j|i,j)|j)) = Var_i(P(i \textrm{ wins }|i)) + \mathbb{E}_i(Var_j(P(i \textrm{ wins } j|i,j)|j))$
>
> The first term on RHS is the classification-based target’s variance and the second term is non-negative. So the classification-based method targets an objective with less noise in it.
>
>
> (2) **Empirically**, such an insight can be observed in experiments. In Figure 7 in Appendix F (page 25), we report the results in barplots on changing the annotation quality.
> - In the top 3 panels, the annotation noises are high (having around or above 30 percent error rate), and in those setups, we find the classification-based reward models in general outperform the BT reward models.
> - In the bottom 3 panels, the annotation noises are low (less than 15 percent error rate), and in those setups, we find the BT models achieve better or on-par performance as compared with the classification-based reward models.

---

> > ### Author Response · Authors · 2024-11-18
> > **Author Response to Reviewer wGfE (Cont.)**
> >
> > ## 3. When cross-prompt comparisons can be particularly effective.
> >
> > We thank the reviewer’s interest in the cross-prompt comparisons. We will try to answer this question from both theoretical and empirical perspectives.
> >
> > (1). **Theoretically**, we have added a section in Appendix G (pages 34-37) that theoretically justifies the superiority of using cross-prompt comparisons in preference annotations. To highlight the high-level takeaways of this section, we show that in expectation, cross-prompt comparisons can improve annotation quality since they increase the expected differences between prompt-response pairs. More specifically, we have
> > - Example 1 and corresponding numerical results to illustrate how annotation quality changes with different response diversity;
> > - Proposition G.1 to show that using cross-prompt comparisons can improve comparison diversity;
> > - Theorem G.2 to show that cross-prompt comparisons can improve annotation quality.
> >
> > In our revised manuscript, those added contents are highlighted with blue text. We additionally used gray text for the proofs to enhance the clarity.
> >
> > (2). **Empirically**, we studied the performances of cross-prompt comparisons under diverse annotation qualities. According to the results in Section 4.3, we can conclude from Figure 4 that cross-prompt comparisons generally improve the alignment performance.
> >
> > In order to understand when those cross-prompt comparisons can be particularly effective, we controlled the generation diversity in response generations. According to the results in Figure 5, we find that **cross-prompt comparison can significantly improve the alignment performance when the responses lack diversity**. To further verify such a finding, we evaluated the correlation between response diversity and the improvement achieved by cross-prompt comparisons (over normal comparisons). We observed a high correlation between those two variables (Figure 6). Those observations validate the insights in the above theory.
> >
> >
> > ---------
> >
> > Once again, we would like to thank the reviewer for their insightful comments and thorough reading to improve our paper. We hope the responses above have addressed the outstanding questions and the reviewer would consider raising their score if those questions have been appropriately answered. Should there be any leftover concerns, please don’t hesitate to let us know and we will do our utmost to address them!

---

> > > ### Comment · Reviewer_wGfE · 2024-11-22
> > >
> > > I appreciate the authors' efforts in addressing the questions. Their response is insightful, supported by both theoretical justification and empirical evidence. Again, these are presented with clarity. To further benefit readers, it would be helpful to include a discussion of Q1 and Q2 in a future revision. Based on these points, I have raised my score.

---

> > > > ### Author Response · Authors · 2024-11-22
> > > > **Thank you for the encouraging feedback!**
> > > >
> > > > Thank you for the encouraging feedback and for raising the score!
> > > > We have included those discussions in our latest revision.
> > > > - The discussion on Q1 has been added to the _Additional Results and Discussions_ section in Appendix F. (please refer to page 24 in the updated manuscript)
> > > > - The discussion on Q2 has been added to the _Theoretical Justifications for Cross-Prompt Comparisons_ section in Appendix G (please refer to pages 34-37 in the updated manuscript)
> > > >
> > > > Once again, we deeply appreciate the reviewer's insightful comments, time, and effort devoted to improving our paper!

---

### Official Review · Reviewer_Ce8Z · 2024-11-04

**Soundness:** 3
**Presentation:** 3
**Contribution:** 3
**Rating:** 8
**Confidence:** 3

**Summary:**

This paper investigates the theoretical validity of employing the BT model in the reward modeling process. It also explores alternative reward modeling approaches and compares them with the BT model. Additionally, this work tests the feasibility and effectiveness of using a cross-prompt comparison method for reward modeling, thereby moving beyond the current reward data paradigm of utilizing the same prompt with different responses.

**Strengths:**

As BT model has been widely used as a  default paradigm in reward modeling, there are few work disscuss the validity for this pattern. This work gives a detailed theoretical analysis of using BT model in reward modeling and offers other alternative methods for reward modeling and  conducts rich ablation studies on them. The theoretical analysis reveals some very basic logics of reward modeling and BT model, which offers a deep explanation of how reward modeling and BT model works.

**Weaknesses:**

1. As illustrated in paper, the label of classifation is simply to  assign (1, 0) to the prefered answer. It is seemingly to be a very rough way for building classifation labels since there are pairs may both be, we say like, 'helpful'. One is labeled as 'chosen' in raw dataset just becase it is more helpful than the rejected one. But it does not mean the rejected one should be  classified in to 'not helpful'.  For example, chosen answer may get a 10/10 score in helpful, and rejected answer may get 9/10.  Can you disscuss this limitation and its potential impact on their results? And have you ever considered alternative labeling schemes that might better capture nuanced differences between responses?
2. The  experiment model size only ranges from 2B to 8B. It remains unknown weather CLF method still more effective when applied to larger models. Do you have any hypotheses about how these results would be if scale to larger models? Can you discuss this limitation and propose how future work could address it?

**Questions:**

1. In the ablation study of Annotation Error Rate, CLF method almost not decrease as Annotation Error Rate arises. This is not common for a  classification model. Can you give more  explanation of this unexpected robustness to annotation errors? And could you also discuss potential implications of this finding for practical applications?
2. Have you considered comparing your methods on existing benchmarks? And how the results in paper might compare to or complement existing benchmark results?

---

> ### Author Response · Authors · 2024-11-18
> **Author Response to Reviewer Ce8Z**
>
> We thank reviewer Ce8Z for their encouraging and insightful comments. We would respond to each of the concerns and questions in turn:
>
> ---
>
> ## Q1. Discussion when fine-grained annotations are available and alternative labeling schemes.
>
> We thank the reviewer for raising the interesting topic of labeling schemes. We agree that a clearer discussion distinguishing our focus on binary annotation from research on alternative labeling schemes would enhance the clarity of our contributions, better position our work, and highlight opportunities for future innovation.
>
> **1. How the cross-prompt comparison becomes helpful in the 10/10 - 9/10 case.**
>
> This is a great concrete example for understanding why the cross-prompt comparison is superior to the single prompt comparison. In practice, it is likely that some of the queries are easier to generate a helpful response (e.g., any generated response is above 7/10) than others (e.g., any generated response is below 3/10 due to harmfulness control). In this case, the annotation quality can be much higher when annotators are asked to compare one response for each prompt rather than making comparisons within the multiple responses having similar scores.
>
> Furthermore, **in Appendix G (pages 34-37) we added a theoretical explanation** of why cross-prompt comparisons can improve annotation quality. In our theoretical development, we characterize quality by the chance of an annotator giving the wrong order. The intuition behind this improvement is that cross-prompt comparisons can improve annotation quality since they increase the expected differences between prompt-response pairs, specifically, we show:
> (1). Example 1 and corresponding numerical results: how annotation quality changes with different response diversity.
> (2). Proposition G.1: using cross-prompt comparisons can improve comparison diversity.
> (3). Theorem G.2: cross-prompt comparisons can improve annotation quality.
>
> Those added contents are highlighted with blue text. To enhance clarity, we additionally used gray text for the proofs.
>
>
> **2. Extended discussion on the literature beyond binary preferences.**
>
> The discussions in our work are developed based on the classical RLHF setup, where annotations are binary preferences over pairs of responses. This line of research is motivated by the assumption that having annotators correctly identify the better one from the worse one can be much easier than having annotators provide precise score information consistently. Under this setup, the BT models are applied to extract scores from those pairwise comparison outcomes.
>
> In addition to building reward models from this binary preference format, there is a line of literature working on exploring other potential rating systems, e.g., having multiple comparisons [1], having rule/guideline-based scoring systems [2,3], or having explanations on the preference annotations using meta-information [4]. The literature uses a data-centric perspective to enhance learning from human feedback research. Exploring the insights discussed in our paper and their variants (e.g., using ordinal regression reward models, or making cross-prompt comparisons to augment the reward modeling dataset) would be promising future directions.
>
> ----
> ## Q2. Scale of model sizes.
>
> We thank the reviewer for raising the question on the model scale.
>
> Our paper focuses on 2B to 8B models, as these lightweight models demonstrate significant potential for applications while offering more tractable inference costs.
>
> From an analytical perspective, we acknowledge that model capabilities improve with scale, potentially leading to performance differences even without alignment. We agree that exploring methods across varying models' abilities is essential to assess the generality of our conclusions.
>
> In our experiments, we used 3 models of different scales, alongside their fine-tuned versions trained on a demonstrative dataset, **as an alternative approach to generating higher-quality responses.** Our experiments consistently show that classification-based reward models outperform others across these setups. For instance, when comparing Gemma2B and Gemma7B base models (Figure 7, Appendix F), their performance gaps are significantly smaller than the gaps between the base models and their SFT-enhanced counterparts. This evidence suggests that SFT influences task performance more than model size, supporting the hypothesis that our findings should extend to larger models.

---

> > ### Author Response · Authors · 2024-11-18
> > **Author Response to Reviewer Ce8Z (Cont.)**
> >
> > ## Q3. explanation of this unexpected robustness to annotation errors, and potential implications of this finding for practical applications
> >
> > We thank our reviewer for raising this insightful point for further discussion.
> >
> > One way to understand the classification-based method is that instead of predicting P(i wins j), we instead predict P(i wins), which is an expectation of P(i wins j) over the randomness of j. One can show that P(i wins) is a lower variance object to be estimated: because it is an average over j of P(i wins j). It can be seen by the law of total variance. Consider the variance
> >
> > $Var_{i,j}(P(i \textrm{ wins } j|i,j)) = Var_i(\mathbb{E}_j(P(i \textrm{ wins } j|i,j)|j)) + \mathbb{E}_i(Var_j(P(i \textrm{ wins } j|i,j)|j)) = Var_i(P(i \textrm{ wins }|i)) + \mathbb{E}_i(Var_j(P(i \textrm{ wins } j|i,j)|j))$
> >
> > The first term on RHS is the classification-based target’s variance and the second term is non-negative. So the classification-based method targets an objective with less noise in it.
> >
> > The discovery in our experiments suggests using classification-based methods when annotation quality is limited and considering using cross-prompt comparisons to improve annotation quality when feasible.
> > On the other hand, when annotation quality is high and the preferences are more deterministic (e.g., a practical case is individualized alignment where all preference labels are from a single user), we should consider using BT models instead of the classification models, since those models, in general, performs better in those cases. (c.f. Figure 2 and Figure 3, when there is less annotation noise, the BT models are better)
> >
> > ----
> >
> > ## Q4. Complement to Existing Benchmarks
> >
> > We thank the reviewer for raising the question on benchmarking the methods.
> >
> > In our paper, we choose to use fixed embeddings as inputs for reward models to isolate the source of gains in reward modeling choices from the representation learning. Our current experiments are designed to verify the insights and theories introduced in the paper and make comparisons between different reward modeling approaches by controlling other variables. Hence our discoveries are more of a complement to existing studies on reward modelling and have the potential to be combined with them.
> >
> > Based on our discoveries, combining the improvement of reward modeling choices and annotation strategies introduced in our work with other orthogonal advances in the literature such as representation learning [5], data merging [6] and regularizations [7] can be important and promising future directions yet out of the current research’s scope.
> >
> >
> >
> > ---
> > **References**
> >
> >
> > [1] Cui, Ganqu, et al. "Ultrafeedback: Boosting language models with high-quality feedback." arXiv preprint arXiv:2310.01377 (2023).
> >
> > [2] Lee, Harrison, et al. "Rlaif: Scaling reinforcement learning from human feedback with ai feedback." arXiv preprint arXiv:2309.00267 (2023).
> >
> > [3] Bai, Yuntao, et al. "Constitutional ai: Harmlessness from ai feedback." arXiv preprint arXiv:2212.08073 (2022).
> >
> > [4] Wu, Tianhao, et al. "Meta-rewarding language models: Self-improving alignment with llm-as-a-meta-judge." arXiv preprint arXiv:2407.19594 (2024).
> >
> >
> > [5] Yang, Rui, et al. "Regularizing Hidden States Enables Learning Generalizable Reward Model for LLMs." arXiv preprint arXiv:2406.10216 (2024).
> >
> > [6] Dong, Hanze, et al. "Rlhf workflow: From reward modeling to online rlhf." arXiv preprint arXiv:2405.07863 (2024).
> >
> > [7] Wang, Binghai, et al. "Secrets of rlhf in large language models part ii: Reward modeling." arXiv preprint arXiv:2401.06080 (2024).
> >
> >
> > --------
> >
> > Once again, we would like to thank the reviewer for their insightful comments and thorough reading to improve our paper. We hope the responses above have addressed the outstanding questions and the reviewer would consider raising their score if those questions have been appropriately answered. Should there be any leftover concerns, please don’t hesitate to let us know and we will do our utmost to address them!

---

> ### Comment · Reviewer_Ce8Z · 2024-11-28
>
> I appreciate your response. The disscussion of cross prompt comparison is convicing. Consequently, I have decided to raise my rating.

---

### Meta-Review · Area_Chair_V67W · 2024-12-18

**Metareview:**

This paper investigates the use of the Bradley-Terry model for reward modeling in LLM alignment, establishing its theoretical foundations while questioning its necessity for downstream optimization. The authors introduce order consistency as a central objective in reward modeling and propose a classification-based alternative that simplifies the process. They further explore cross-prompt comparisons to enhance annotation quality and model performance. The work is supported by extensive experiments across diverse setups, providing actionable insights for reward modeling.

Reviewers unanimously praised the paper’s solid theoretical contributions, extensive empirical evaluation, and clear presentation. Key strengths include the introduction of order consistency as a fundamental principle, the innovative classification-based approach, and the effective use of cross-prompt comparisons to improve annotation quality.

**Additional Comments On Reviewer Discussion:**

During the rebuttal, reviewers raised concerns about the limited model size range (2B–8B) and sought clarification on the robustness of classification-based methods and the benefits of cross-prompt comparisons. The authors addressed these concerns comprehensively. They also included detailed discussions on scenarios where cross-prompt comparisons are most effective. These changes and clarifications resolved all major concerns.

---

### Decision · Program_Chairs · 2025-01-22

Accept (Oral)